# SONA: Learning Conditional, Unconditional, and Matching-Aware Discriminator

**Yuhta Takida**[1] **Satoshi Hayakawa**[2] **Takashi Shibuya**[1] **Masaaki Imaizumi**[3]
**Naoki Murata**[1] **Bac Nguyen**[1] **Toshimitsu Uesaka**[1] **Chieh-Hsin Lai**[1] **Yuki Mitsufuji**[1,2]
[1]Sony AI,  [2]Sony Group Corporation,  [3]The University of Tokyo

## Abstract

Deep generative models have made significant advances in generating complex content, yet conditional generation remains a fundamental challenge. Existing conditional generative adversarial networks often struggle to balance the dual objectives of assessing authenticity and conditional alignment of input samples within their conditional discriminators. To address this, we propose a novel discriminator design that integrates three key capabilities: unconditional discrimination, matching-aware supervision to enhance alignment sensitivity, and adaptive weighting to dynamically balance all objectives. Specifically, we introduce **S**um **of** **N**aturalness and **A**lignment (SONA), which employs separate projections for naturalness (authenticity) and alignment in the final layer with an inductive bias, supported by dedicated objective functions and an adaptive weighting mechanism. Extensive experiments on class-conditional generation tasks show that SONA achieves superior sample quality and conditional alignment compared to state-of-the-art methods. Furthermore, we demonstrate its effectiveness in text-to-image generation, confirming the versatility and robustness of our approach.

## 1 Introduction

Deep generative modeling has achieved remarkable progress in synthesizing images (Podell et al., 2024; Esser et al., 2024), audio (Novack et al., 2024; 2025), and video (Yang et al., 2025; Polyak et al., 2024; Kong et al., 2024; Wan et al., 2025). Nevertheless, generating high-quality samples that are well-aligned with conditional information, such as class labels or text prompts, remains a central challenge (Ho & Salimans, 2021; Dhariwal & Nichol, 2021; Liu et al., 2023; Zhang et al., 2024).

Generative adversarial networks (GANs) (Goodfellow et al., 2014) have been instrumental in advancing conditional generation, with much of the research focusing on the design of conditional discriminators (Kang et al., 2023a). The task of the conditional discriminator is typically decomposed into two sub-problems: distinguishing real from generated samples (unconditional discrimination) and assessing conditional alignment. This decomposition can be naturally motivated by the likelihood factorization of the joint distribution, $p(x, y) = p(y|x)p(x)$, where $x$ is a data sample and $y$ is a conditioning variable.

Two main approaches have emerged based on this factorization. The ***classifier-based*** approach, pioneered by AC-GAN (Odena et al., 2017), uses a dual-head discriminator to simultaneously evaluate sample authenticity and label alignment (Gong et al., 2019; Hou et al., 2022; Kang et al., 2021). The ***projection-based*** approach, introduced by Miyato & Koyama (2018), models the discriminator as a sum of unconditional discrimination and alignment terms, eliminating the need for auxiliary classifiers and thereby simplifying the architecture. This simple yet effective design has been widely adopted in modern conditional GANs as the de facto standard without major modifications (Brock et al., 2019; Karras et al., 2019; 2020b; 2021; Sauer et al., 2022; 2023; Huang et al., 2024).

Despite these advances, conditional discriminators still face the fundamental challenge of balancing the dual objectives of unconditional discrimination and conditional alignment (Reed et al., 2016; Hou et al., 2022). Classifier-based methods require careful tuning of weighting coefficients to achieve this balance (Kang et al., 2021). We also suspect that projection-based methods may not

Table 1: Three desiderata for our proposed method, SONA.

| Capabilities | Conditional discrimination | (i) Unconditional disc. (Section 3.1) | (ii) Matching-aware disc. (Section 3.2) | (iii) Adaptive weighting (Section 3.3) |
|---|---|---|---|---|
| Classifier-based | ✓ | ✓ | ✓ | |
| Projection-based | ✓ | * | | N/A |
| SONA (ours) | ✓ | ✓ (Section 4.2) | ✓ (Section 4.3) | ✓ (Section 4.4) |

fully exploit the likelihood decomposition for the unconditional discrimination task, as discussed in Section 3.

To address this issue, we aim at a discriminator design that incorporates three capabilities, as summarized in Table 1. First, we introduce *(i) unconditional discrimination* to robustly distinguish real from fake samples, independent of the condition. Second, we enhance the discriminator's sensitivity to conditional alignment by providing additional supervision through mismatched (negative) samples, resulting in a *(ii) matching-aware discriminator* (Reed et al., 2016; Zhang et al., 2017; Tao et al., 2023; Kang et al., 2023b). Third, we employ an *(iii) adaptive weighting* mechanism to dynamically balance the objectives of conditional, unconditional, and matching-aware discrimination.

Specifically, we introduce Sum of Naturalness and Alignment (SONA), a novel method that simultaneously fulfills all the capabilities listed in Table 1, as detailed in Section 4. Our discriminator is designed with separate projections to independently assess input naturalness (authenticity) and conditional alignment, while incorporating an effective inductive bias to support both tasks efficiently. To fully leverage this architecture, we propose a set of objective functions for training conditional, unconditional, and matching-aware discrimination, and validate their effectiveness both theoretically and empirically in Section 5. Additionally, we introduce a simple yet effective adaptive weighting mechanism for these three discrimination tasks, enabled by our carefully designed loss functions. In Section 6, we evaluate SONA on image datasets with class labels, demonstrating that it generates higher-quality samples with better conditional alignment than state-of-the-art (SoTA) discriminator conditioning methods. We further extend our experiments to text-to-image generation, showing the applicability of SONA to more complex conditioning scenarios.

## 2 PRELIMINARIES

Let $p_{\text{data}}(x, y)$ represent the data distribution, where $x \in X$ is a data sample and $y \in Y$ is the conditional information describing the corresponding $x$ (e.g., a class label or text prompt). Our objective is to learn the conditional distribution from a finite set of samples drawn from it. For this purpose, a trainable generator is introduced, denoted as $g$, inducing the generator distribution, denoted as $p_g(x|y)$. In one of the standard GAN setups, the generator is parameterized as a function that transforms a tractable noise (e.g., a Gaussian noise) to a data sample as $g_\theta : Z \times Y \to X$, where $\theta$ indicates a set of parameters modeling the generator and $z \in Z$ is the noise; thus a sample drawn from $p_g(\cdot|y)$ is obtained by $x_g = g_\theta(z, y)$ with a noise drawn from a base distribution: $z \sim p_Z$. Hereafter, we use $\mathcal{V}$ and $\mathcal{J}$ to denote maximization and minimization objectives, respectively.

### 2.1 GENERATIVE ADVERSARIAL NETWORKS

We review the formulation of GANs and introduce the sliced Wasserstein perspective to present the concept of optimal projection for unconditional discrimination. The problem setup described above includes unconditional generation tasks by setting $y$ to null conditioning. In this subsection, we omit $y$ from the formulations for simplicity.

**GANs.** In GANs, a discriminator, denoted as $f : X \to \mathbb{R}$, is introduced, which is expected to discriminate between the samples drawn from the data and generator distributions with its scalar outputs. GAN formulates the optimization problem to make the generator distribution closer to the data distribution by solving a minimax problem:

$$\max_f \mathcal{V}_{\text{GAN}}(f; g), \quad \text{and} \quad \min_g \mathcal{J}_{\text{GAN}}(g; f). \tag{1}$$

Here, the variables following the semicolons are held fixed during each optimization step, and we will omit such variables when the context is clear. The specific forms of $\mathcal{V}_{\text{GAN}}$ and $\mathcal{J}_{\text{GAN}}$ depend on the chosen GAN variant or loss (see Appendix B.1 for more details).

**Sliced Wasserstein perspective on GANs.** Typical discriminators can be represented as $f(x) = \langle \omega, h(x) \rangle$, where $h : X \to \mathbb{R}^D$, $\omega \in \mathbb{S}^{D-1}$, and $\langle \cdot, \cdot \rangle$ denotes the Euclidean inner product. Takida et al. (2024) interpreted this formulation as an augmented Sliced Wasserstein approach (Kolouri et al., 2019; Chen et al., 2022) with a single direction ($\omega$). Building on this interpretation, they propose encouraging optimality in the sliced Wasserstein sense on the normalized projection, resulting in slicing adversarial networks (SANs): $\max_{\omega,h} \mathcal{V}_{\text{SAN}}(\omega, h)$ and $\min_g \mathcal{J}_{\text{SAN}}(g)$, where

$$\mathcal{V}_{\text{SAN}}(\omega, h) = \mathbb{E}_{p_{\text{data}}(x)}[\langle \omega, \mathtt{sg}(h)(x) \rangle] - \mathbb{E}_{p_g(x)}[\langle \omega, \mathtt{sg}(h)(x) \rangle] + \mathcal{V}_{\text{GAN}}(\langle \mathtt{sg}(\omega), h \rangle), \quad (2)$$

$$\mathcal{J}_{\text{SAN}}(g) = -\mathbb{E}_{p_g(x)}[\langle \omega, h(x) \rangle], \quad (3)$$

where $\mathtt{sg}(\cdot)$ denotes the stop-gradient operator[1]. The first two terms in $\mathcal{V}_{\text{SAN}}$ encourage the direction $\omega$ to maximize the sliced Wasserstein distance given by $h$. Intuitively, the learned direction is expected to optimally distinguish real and generated samples in the feature space defined by $h$. See Appendix B.2 for a more detailed explanation of the aforementioned perspective.

## 2.2 Conditional GANs

Most conditional GANs employ either classifier-based or projection-based approaches. To illustrate the core concepts, we briefly review AC-GAN as a representative classifier-based approach, as well as the projection-based approach. A detailed review of related work is provided in Appendix A.

In conditional generation settings, the discriminator is modeled as $f : X \times Y \to \mathbb{R}$, enabling it to distinguish between the two conditional distributions, $p_{\text{data}}(x|y)$ and $p_g(x|y)$. For simplicity, we assume $Y$ is a discrete space in this subsection.

**Classifier-based approach.** Odena et al. (2017) introduced AC-GAN, which combines the original GAN losses (i.e., $\mathcal{V}_{\text{GAN}}$ and $\mathcal{J}_{\text{GAN}}$) with cross-entropy classification losses: $\mathcal{V}_{\text{CLS}} = \mathbb{E}_{p_{\text{data}}(x,y)}[\log C(x, y)]$ and $\mathcal{J}_{\text{CLS}} = -\mathbb{E}_{p_g(x,y)}[\log C(x, y)]$ to optimize the discriminator and generator. The auxiliary classifier is typically defined as $C(x, y) = \text{softmax}_y(\{\tilde{f}_{\text{cls}}(x, y)\}_{y \in Y}/\tau)$, where $\tilde{f}_{\text{cls}} : X \times Y \to \mathbb{R}$ and $\tau \in \mathbb{R}_{>0}$ is a temperature. Notably, under this setup, the maximization loss $\mathcal{V}_{\text{CLS}}$ is equivalent to the InfoNCE loss (Oord et al., 2018):

$$\mathcal{V}_{\text{CE}}(\tilde{f}_{\text{cls}}) = \mathbb{E}_{p_{\text{data}}(x,y)} \left[ \log \frac{\exp(\tilde{f}_{\text{cls}}(x, y)/\tau)}{\mathbb{E}_{p_{\text{data}}(y')} \exp(\tilde{f}_{\text{cls}}(x, y')/\tau)} \right]. \quad (4)$$

To enable the discriminator to predict class labels, $\tilde{f}_{\text{cls}}(x, y)$ is further parameterized using the discriminator's deep feature and additional learnable embeddings $w_y \in \mathbb{R}^D$ as $\tilde{f}_{\text{cls}}(x, y) = \langle w_y, h(x) \rangle$.

**Projection-based approach.** Miyato & Koyama (2018) proposed a simple yet effective discriminator design. Based on the Bayes-rule-based log-likelihood-ratio factorization, they implement the discriminator as a sum of conditional and unconditional terms: $f(x, y) = f_1(x, y) + f_2(x) = \langle w_y, h(x) \rangle + \psi(h(x))$, where, by abuse of notation, $w_y$ denotes the embedding of $y$, and $\psi$ is a learnable function. For efficient optimization, the intermediate feature $h(x)$ is shared between $f_1(x, y)$ and $f_2(x)$. In practice, $\psi$ is usually parameterized as a linear layer, reducing the discriminator to

$$f(x, y) = \langle w_y, h(x) \rangle + \langle w, h(x) \rangle + b, \quad (5)$$

where $b \in \mathbb{R}$ is a learnable bias. This approach does not require any modifications other than the projection discriminator, such as optimization schemes or objectives. It is widely used in its original form, and we hereafter refer to the broad class of GANs based on this approach simply as PD-GANs.

## 3 Motivation: Key Capabilities of Conditional Discriminator

In this section, we raise the desirable capabilities for our discriminator, and discuss whether the existing classifier- and projection-based approaches satisfy these criteria (see Table 1 for a summary).

---

[1] For any function $U : X \to \mathbb{R}$, $\mathbb{E}_{p_g(x)}[U(x)]$ is equivalent to $\mathbb{E}_{p_Z(z)}[U(g(y, z))]$, and is therefore differentiable with respect to $g$. We adopt the former notation for simplicity throughout this manuscript.

## 3.1 Unconditional Discrimination

We argue that unconditional discriminator learning is essential even in conditional generation tasks, as existing approaches decompose the role of the conditional discriminator into unconditional discrimination and evaluation of conditional alignment. Classifier-based discriminators inherently support unconditional discrimination by explicitly employing the unconditional GAN loss (see Section 2.2). In contrast, projection-based discriminators, when used with standard GAN losses, may not provide this capability, as discussed below.

Projection-based discriminators are equipped with both unconditional and conditional projections, as shown in Equation (5), and are thus inherently capable of modeling unconditional discrimination. However, since Equation (5) can be rewritten as $\langle \tilde{w}_y, h(x) \rangle + b$ with $\tilde{w}_y = w_y + w \in \mathbb{R}^D$, the generator is optimized by $\min_g \mathcal{J}_{\text{GAN}}(g; \langle \tilde{w}_y, h(x) \rangle + b)$, essentially with a $y$-dependent projection $\tilde{w}_y$. This suggests that, even with this parameterization, the objective functions typically used in PD-GANs may not fully leverage unconditional discrimination.

## 3.2 Matching-Aware Discrimination

We next highlight the importance of enhancing the discriminator's sensitivity to conditional alignment by incorporating negative samples, following the approach of Reed et al. (2016). Specifically, to encourage conditional alignment, they proposed using negative samples that are realistic but associated with incorrect class labels, thereby mismatching the conditional information.

As shown in Equation (4), AC-GAN can be interpreted as implicitly utilizing such negative samples drawn from the product of marginals, i.e., $(x, y') \sim p_{\text{data}}(x)p_{\text{data}}(y')$, in its cross-entropy loss, in addition to samples from the true joint distribution $p_{\text{data}}(x, y)$. This advantage is formalized in Proposition 1, which implies that the cross-entropy loss induces the discriminator feature to be $y$-extractable, sensitive to conditional alignment, under the assumption that $p_{\text{data}}(y)$ is uniform. This proposition imposes the uniform assumption on $p_{\text{data}}(y)$, which holds for well-constructed image datasets (Krizhevsky et al., 2009; Russakovsky et al., 2015), where each class contains the same number of samples.

**Proposition 1** (Log conditional probability maximizes $\mathcal{V}_{\text{CE}}$). *Assume $p_{\text{data}}(y)$ is a constant regardless of $y \in Y$, e.g., a uniform distribution. The function $\tilde{f}$ maximizes $\mathcal{V}_{\text{CE}}$ if $\tilde{f}(x, y) = \log p_{\text{data}}(y|x) + r_X(x)$ for an arbitrary function $r_X : X \to \mathbb{R}$.*

While classifier-based approaches (including but not limited to AC-GAN) employ classification losses similar or analogous to InfoNCE, projection-based GANs do not incorporate such losses, resulting in the absence of explicit mechanisms for inducing matching-awareness.

## 3.3 Desiderata of Our Discriminator

Classifier-based GANs possess the two additional discrimination capabilities outlined in the previous subsections, while most PD-GANs do not. However, a key advantage of PD-GANs is that they avoid introducing additional hyperparameters that require manual tuning, which is beneficial for practitioners. In contrast, our goal is to propose a novel conditional discriminator that integrates both unconditional and matching-aware discrimination into the training process, while adaptively balancing these different objectives. A summary of these comparisons is shown in Table 1.

## 4 Proposed Method: SONA

To achieve the desiderata presented in Section 3, we design the discriminator and propose a set of maximization objective functions for it. We provide theoretical and empirical support for our method in Section 5. We formalize the training procedure of SONA in Algorithm 1 of Appendix C.

## 4.1 Discriminator Parametrization

Inspired by the projection discriminator (Equation (5)), we design the discriminator to evaluate sample inputs by summing two scalar terms for (a) the (unconditional) naturalness, i.e., distinguishing

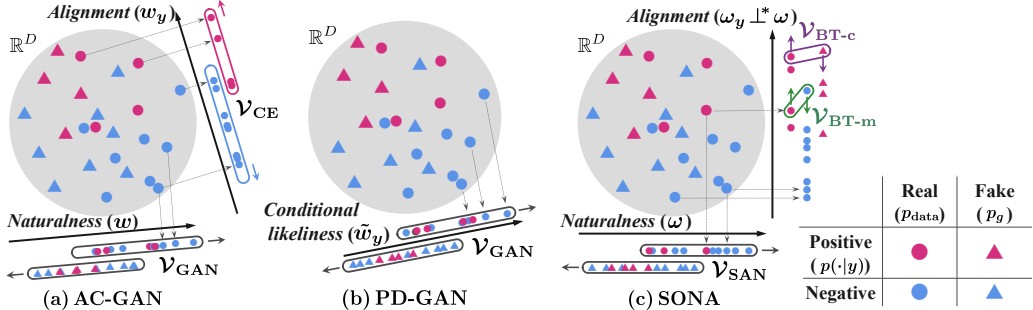

Figure 1: Comparison of SONA with existing classifier- and projection-based methods for discriminator optimization. Our approach enables independent assessment of sample naturalness and alignment, supported by the proposed inductive bias (Section 4.1) and objectives (Sections 4.2 and 4.3).

real from fake samples, and (b) the alignment with the conditioning information. To achieve this compositional modeling, we introduce a feature extractor $h : X \to \mathbb{R}^D$, shared across both tasks (here, $h$ is consistent with the notation in Section 2). The extracted features $h(x)$ are then projected onto independent directions $\omega \in \mathbb{S}^{D-1}$ and $\omega_y \in \mathbb{S}^{D-1}$ for each $y \in Y$ as follows.

For naturalness, we simply project the feature onto $\omega$. For conditional alignment, we incorporate an inductive bias based on the hypothesis that assessing naturalness and conditional alignment are orthogonal tasks. From an optimization perspective, optimizing the generator for alignment should not interfere with optimizing it for naturalness. To encode this inductive bias, we define the alignment term using an orthogonal projection: $\langle \omega_y, \Pi_{\perp\omega} h(x) \rangle$, where $\Pi_{\perp\omega} h(x) = h(x) - \langle \omega, h(x) \rangle \omega$.

Thus, our discriminator is parameterized as the sum of these two terms

$$f(x,y) = \underbrace{\langle \omega, h(x) \rangle}_{f_{\Phi_N}^N(x):\ \underline{\text{N}}\text{aturalness}} + \underbrace{\langle \omega_y, \Pi_{\perp\omega} h(x) \rangle}_{f_{\Phi_A}^A(x,y):\ \text{conditional } \underline{\text{A}}\text{lignment}}, \tag{6}$$

where $\Phi_N = \{\omega, h\}$, $\Phi_A = \{\omega, \omega_y, h\}$. In this formulation, we expect $\omega$ to be responsible for distinguishing the naturalness of input samples (as in Section 4.2), while $\omega_y$ focuses on conditional alignment (as in Section 4.3). Here, we can optionally add a bias $b \in \mathbb{R}$ to the naturalness term, which can also be absorbed into $h$. Please also refer to Figure 1 for an illustration of our strategy.

## 4.2 UNCONDITIONAL LEARNING

To address **the first desideratum** in Table 1, we formulate a minimax problem that encourages the naturalness term in Equation (6) to distinguish between real and generated samples independently of $y$ (see Proposition 2 in Section 5.1). We employ SAN objective functions to learn the optimal $\omega$ for unconditional discrimination, specifically using Equations (2) and (3) as the minimax objectives:

$$\max_{\Phi_N} \mathcal{V}_{\text{SAN}}(\omega, h), \quad \text{and} \quad \min_g \mathcal{J}_{\text{SAN}}(g). \tag{7}$$

Note that only the parameters associated with the naturalness term $f_N$ in Equation (6) are included; $\Phi_A \setminus \Phi_N = \{\omega_y\}$, which is used only for conditional alignment, is not involved. This optimization ensures that $\omega$ focuses on determining whether an input sample originates from the data or the generator, as intended. We denote $\mathcal{V}_{\text{SAN}}(\omega, h)$ as $\mathcal{V}_{\text{SAN}}(\Phi_N)$ in Equation (12) for a unified formulation.

## 4.3 LEARNING CONDITIONAL ALIGNMENT

Next, we develop $\omega_y$-based learning for the conditional alignment, building on the $\omega$-based unconditional learning described in Section 4.2. Specifically, we introduce additional objective terms to enable our discriminator to perform conditional discrimination and to be aware of the mismatch, the latter corresponding to **the second desideratum** in Table 1.

To achieve this, we incorporate the Bradley–Terry (BT) model (Bradley & Terry, 1952, reviewed in Appendix B.3), which is widely recognized for its efficiency in modeling pairwise comparisons and

has recently been applied in reinforcement learning from human feedback (Rafailov et al., 2023). For each pair of samples, we denote the preferred sample as the "winning" sample $x_w$ and the less preferred as the "losing" sample $x_\ell$, response for condition $y$. The model defines the probability that $x_w$ is preferred over $x_\ell$ given $y$ using an evaluation function $\tilde{f} : X \times Y \to \mathbb{R}$ as $\Pr(x_w \text{ is preferred over } x_\ell | y) = \sigma(\tilde{f}(x_w, y) - \tilde{f}(x_\ell, y))$, where $\sigma(\cdot)$ denotes the sigmoid function. Following standard practice, we optimize our discriminator $f$ by maximizing the following likelihood:

$$\mathcal{V}_{\mathrm{BT}} = \mathbb{E}_{x_w, x_\ell, y}[\log \sigma(f(x_w, y) - f(x_\ell, y))]. \tag{8}$$

In our framework, samples drawn from the true joint distribution $p_{\mathrm{data}}(x, y)$ are always designated as the winning samples $x_w$, since this distribution represents the target. For the losing samples $x_\ell$, we consider two distinct distributions, resulting in two additional objectives, as follows.

**BT-C loss for conditional discrimination.** The first losing distribution is the generator distribution. From Equation (8), the corresponding BT loss is

$$\mathcal{V}_{\mathrm{BT\text{-}C}}(f_{\Phi_A}^A) = \mathbb{E}_{p_{\mathrm{data}}(y)p_{\mathrm{data}}(x_w|y)p_g(x_\ell|y)}[\log \sigma(f_{\mathrm{sg}(\Phi_N)}^N(x_w) + f_{\Phi_A}^A(x_w, y) - f_{\mathrm{sg}(\Phi_N)}^N(x_\ell) - f_{\Phi_A}^A(x_\ell, y))]. \tag{9}$$

This BT loss compares real and generated samples conditioned on a given $y$, thereby measuring conditional dissimilarity. The sum of the first two terms corresponds to $f(x_w, y)$, while the latter two correspond to $f(x_\ell, y)$. Notably, since the objective her is to learn conditional alignment, the parameters $\Phi_A$ are optimized only through the alignment term $f_{\Phi_A}^A$, while the naturalness term $f_{\Phi_N}^N$ is frozen by applying the stop-gradient operator solely to the naturalness term. Under optimality assumptions, including those related to Equation (9), Equation (9) can be interpreted as a specific divergence between $p_{\mathrm{data}}(x|y)$ and $p_g(x|y)$, up to constant, as shown in Proposition 3 of Section 5.1.

**BT-M loss for matching-aware discrimination.** The second losing distribution, chosen to address **the second desideratum**, is the marginal data distribution, which ignores the given condition $y$. This helps the discriminator identify samples that do not satisfy the specified condition, even if they are real samples. The corresponding BT loss is

$$\mathcal{V}_{\mathrm{BT\text{-}M}}(f_{\Phi_A}^A) = \mathbb{E}_{p_{\mathrm{data}}(y)p_{\mathrm{data}}(x_w|y)p_{\mathrm{data}}(x_\ell)}[\log \sigma(f_{\mathrm{sg}(\Phi_N)}^N(x_w) + f_{\Phi_A}^A(x_w, y) - f_{\mathrm{sg}(\Phi_N)}^N(x_\ell) - f_{\Phi_A}^A(x_\ell, y))]. \tag{10}$$

This BT loss compares data samples aligned with the condition $y$ against negative samples drawn from the marginal distribution, analogous to a matching loss. As shown in Proposition 4 of Section 5.1, maximizing Equation (10) with respect to the discriminator yields the log gap between the conditional and unconditional probabilities, $\log p_{\mathrm{data}}(x|y) - \log p_{\mathrm{data}}(x)$, which is useful for enhancing conditional alignment (Ho & Salimans, 2021; Chen et al., 2025b).

**Minimization optimization for conditional alignment.** Finally, we introduce a minimization objective for generator optimization with respect to conditional alignment. By swapping the data and generator distributions in Equation (9), we obtain a minimization loss analogous to that used in relativistic pairing GAN (Jolicoeur-Martineau, 2018):

$$\mathcal{J}_{\mathrm{BT\text{-}C}}(g) = -\mathbb{E}_{p_{\mathrm{data}}(y)p_g(x_g)p_{\mathrm{data}}(x_d|y)}[\log \sigma(f_{\Phi_N}^N(x_{\mathrm{sg}(g)}) + f_{\Phi_A}^A(x_g, y) - f_{\Phi_N}^N(x_d) - f_{\Phi_A}^A(x_d, y))], \tag{11}$$

Here, a slight modification is added: as in Equations (9) and (10), the stop-gradient operator is applied only in the naturalness term (note that the third term does not include $g$), ensuring that minimization occurs orthogonally to the direction represented by $\omega$ (see the orthogonal operator in Equation (6)). This approach allows the loss to specifically enhance the conditional alignment of generated samples along $\omega_y$, while authenticity is enforced by $\mathcal{J}_{\mathrm{SAN}}$ using the direction responsible for unconditional discrimination. Therefore, minimizing $\mathcal{J}_{\mathrm{SAN}}$ and $\mathcal{J}_{\mathrm{BT\text{-}C}}$ does not cause interference, enabling each objective to address its respective aspect independently.

## 4.4 OVERALL OBJECTIVE FUNCTION WITH ADAPTIVE WEIGHTING

We have introduced the maximization and minimization objective terms in Sections 4.2 and 4.3. The overall objective for training our GAN is summarized as follows:

$$\max_{\Phi_N \cup \Phi_A} \mathcal{V}_{\mathrm{SAN}}(\Phi_N) + \mathcal{V}_{\mathrm{BT\text{-}C}}(f_{\Phi_A}^A) + \mathcal{V}_{\mathrm{BT\text{-}M}}(f_{\Phi_A}^A), \text{ and } \min_g \mathcal{J}_{\mathrm{SAN}}(g) + \mathcal{J}_{\mathrm{BT\text{-}C}}(g). \tag{12}$$

To ensure adaptive balance among the maximization objective terms $\mathcal{V}_{\text{SAN}}$, $\mathcal{V}_{\text{BT-C}}$, and $\mathcal{V}_{\text{BT-M}}$, we introduce learnable scalar parameters. Specifically, we first adopt $\mathcal{V}_{\text{GAN}}$ from Goodfellow et al. (2014) to construct $\mathcal{V}_{\text{SAN}}$, which is formulated with $\log \sigma(\cdot)$ (see Appendix B.1). We then replace $\log \sigma(t)$ in each of $\mathcal{V}_{\text{SAN}}$, $\mathcal{V}_{\text{BT-C}}$, and $\mathcal{V}_{\text{BT-M}}$ with $\log \sigma(s \cdot t)/s$, where $s \in \mathbb{R}_{>0}$ is learnable. To prevent these coefficients from diverging, we constrain them such that $s_{\text{SAN}}^2 + s_{\text{BT-C}}^2 + s_{\text{BT-M}}^2 = 1$. This approach makes the adaptive weighting possible by incorporating the current situation during training (see Appendix E.4 for details), thereby satisfying **the third desideratum** in Table 1.

Adaptive weighting has been investigated in general multi-task learning (Kendall et al., 2018). However, these approaches are not specifically designed for GAN training, so they may not be suitable for our purpose. In particular, we hypothesize that the unbounded nature of the coefficients in these approaches can be harmful to GAN training, as GANs are highly sensitive to the learning rate (Heusel et al., 2017). This hypothesis motivates the development of our adaptive weighting mechanism. To empirically validate our hypothesis and demonstrate the effectiveness of our method, we compare it with the approach proposed by Kendall et al. (2018), using the same experimental setup described in Section 6.3. Our method achieves an FID of 5.65±0.25 and an IS of 9.51±0.05, which are significantly better than the baseline results of FID 16.62±4.04 and IS 7.88±0.80.

# 5 ANALYSIS OF SONA

## 5.1 THEORETICAL GROUNDING FOR OUR MAXIMIZATION OBJECTIVES

In this subsection, we present propositions to demonstrate the validity of the objective terms introduced in Section 4.2 and Section 4.3. Proofs are provided in the Appendix.

First, the following proposition, which is a restatement of Theorem 5.3 in Takida et al. (2024), establishes that optimizing the generator and discriminator using the minimax objective functions from Section 4.2 enables unconditional GAN learning.

**Proposition 2** (Informal; Unconditional discrimination by $\mathcal{V}_{\text{SAN}}$). *Let the unconditional discriminator (the naturalness term) be $f^{\text{N}}(x) = \langle \omega, h(x) \rangle$ with $\omega \in \mathbb{R}^{D-1}$ and $h : X \to \mathbb{R}^D$. Under suitable regularity conditions for $h$, the objective $\mathcal{J}_{\text{SAN}}(g; \hat{\omega}, h)$ is minimized only if $g$ minimizes a certain distance between $p_{\text{data}}(x)$ and $p_g(x)$, where $\hat{\omega} = \arg\max_{\omega} \mathcal{V}_{\text{SAN}}(\omega, h)$ for a given $h$.*

Next, we analyze the BT-based objective functions introduced in Section 4.3. BT-C loss $\mathcal{V}_{\text{BT-C}}$ compares samples from the dataset and the generator with specific conditioning. Under certain optimal conditions, this loss can represent the conditional dissimilarity between conditional distributions, as demonstrated in Proposition 3.

**Proposition 3** (Conditional discrimiantion by $\mathcal{V}_{\text{BT-C}}$). *Let the discriminator be $f(x, y) = f^{\text{N}}(x) + f^{\text{A}}(x, y)$, where $f^{\text{N}}(x) = \langle \omega, h(x) \rangle$ with $\omega \in \mathbb{S}^{D-1}$, $h : X \to \mathbb{R}^D$, $b \in \mathbb{R}$, and $f^{\text{A}} : X \times Y \to \mathbb{R}$. Assume that the generator achieves $p_g(x) = p_{\text{data}}(x)$, and $\omega$ and $h$ maximize $\mathcal{V}_{\text{SAN}}$ for given $p_{\text{data}}$ and $p_g$. If $f^{\text{A}}$ maximizes Equation (13), then it is minimized if and only if $p_{\text{data}}(x|y) = p_g(x|y)$ for $y \in Y$.*

$$\mathcal{L}_{\text{BT-C}} = \mathbb{E}_{p_{\text{data}}(y)p_{\text{data}}(x_w|y)p_g(x_\ell|y)}[\log \sigma(f^{\text{N}}(x_w) + f^{\text{A}}(x_w, y) - f^{\text{N}}(x_\ell) - f^{\text{A}}(x_\ell, y))]. \quad (13)$$

Here, Equation (13) corresponds to the RHS of Equation (9) with generalized terms. We note that in our method, the conditional alignment term ($f^{\text{A}}$ in this proposition) shares $h$ and $\omega$ with the naturalness term, a constraint not considered in this proposition. However, since minimizing $\mathcal{V}_{\text{SAN}}$ enforces only one-dimensional constraint on $h$ given $\omega$, we expect $f^{\text{A}}_{\Phi_{\text{A}}}$ to have sufficient capacity even conditioned on $\mathcal{V}_{\text{SAN}}$-minimization. Thus, this proposition still offers valuable insights.

Finally, BT-M loss using samples from the marginal data distribution, $\mathcal{V}_{\text{BT-M}}$, can be interpreted as a contrastive loss comparing positive and negative data samples. Specifically, $\mathcal{V}_{\text{BT-M}}$ is equivalent to an InfoNCE loss with a single negative sample per positive sample in its denominator. The following proposition shows that this objective encourages the conditional discriminator to learn the log gap between conditional and unconditional probabilities, up to an arbitrary function independent of $x$:

**Proposition 4** (Log gap probability maximizes $\mathcal{V}_{\text{BT-M}}$). *The function $\tilde{f}$ maximizes $\mathcal{V}_{\text{BT-M}}$ if $\tilde{f}(x, y) = \log p_{\text{data}}(x|y) - \log p_{\text{data}}(x) + r_Y(y)$ for an arbitrary function $r_Y : Y \to \mathbb{R}$.*

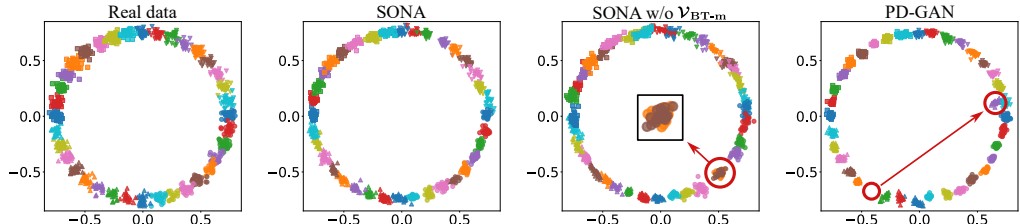

Figure 2: Empirical study on MoG using Wasserstein-2 distance (**W2**), Conditional Wasserstein-2 distance (**cW2**), and the number of failure cases (**NF**). See Section 5.2 and Appendix F.1 for details.

Figure 3: Ground truth samples and generated samples from three baseline models. Different markers and colors represent samples from distinct classes among the $N = 36$ total classes.

Although Proposition 4 superficially resembles Proposition 1 in Section 3.2, there are two key differences. First, Proposition 4 does not require the uniform assumption on $p_{\text{data}}(y)$, allowing it to be applied to broader settings, such as datasets with biased class distributions or text-caption–image datasets. Second, the extra term in the maximizer of Proposition 4 is independent of $x$, unlike in Section 3.2. This means that the maximizer captures the score gap between the conditional and unconditional probabilities, which helps to emphasize conditional alignment.

## 5.2 EMPIRICAL VALIDATION OF OUR METHOD

To empirically evaluate the effectiveness of our proposed method, we conduct experiments on a two-dimensional mixture of Gaussians (MoG) dataset, which enables both visualization and accurate measurement of generative performance. The experimental details are provided in Appendix F.1.

We train three models on the MoG dataset, varying the number of Gaussians (i.e., classes), denoted as $N$: (1) SONA, (2) SONA without the matching loss $\mathcal{V}_{\text{BT-M}}$, and (3) PD-GAN. To quantitatively assess generative performance, we use three metrics: (a) Wasserstein-2 distance ($\text{W}_2(p_{\text{data}}(x), p_g(x))$, denoted as **W2**), (b) conditional Wasserstein-2 distance ($\frac{1}{N}\sum_{n=1}^{N} \text{W}_2(p_{\text{data}}(x|y_n), p_g(x|y_n))$, denoted as **cW2**), and (c) the number of failure cases (**NF**). A failure is counted if there exists $n \in [N]$ such that $\text{W}_2(p_{\text{data}}(x|y_n), p_g(x|y_n)) > \epsilon$, where $\epsilon$ is set to the standard deviation of the Gaussians.

As shown in Figure 2, using five different random seeds, generators trained with SONA demonstrate robust performance, consistently outperforming the baselines when $N \geq 30$. Notably, SONA achieves zero **NF**, while the other two methods increasingly fail as $N$ grows. Qualitative results for $N = 36$ are visualized in Figure 3, where PD-GAN fails to cover all modes. In contrast, SONA without the matching loss produces overlapping samples between classes, indicating difficulty in distinguishing between them. This underscores the importance of making the discriminator matching-aware to better utilize conditional information (**the second desideratum**).

## 6 EXPERIMENTS

### 6.1 BENCHMARK ON CLASS-CONDITIONAL GENERATION TASKS

We conduct class-conditional image generation experiments on CIFAR10 (Krizhevsky et al., 2009), TinyImageNet (Le & Yang, 2015), and ImageNet (Deng et al., 2009), using the StudioGAN repository (Kang et al., 2023a)[2], a well-established benchmark for these tasks. As baselines, we select two state-of-the-art (SoTA) classifier-based methods, ReACGAN (Kang et al., 2021) and Contra-GAN (Kang & Park, 2020), as well as PD-GANs, which is among the most widely used approaches.

---

[2] https://github.com/POSTECH-CVLab/PyTorch-StudioGAN

Table 2: CIFAR10

| Method | FID ↓ | IS ↑ |
|---|---|---|
| *BigGAN backbone* | | |
| ContraGAN | 4.74±0.05 | 9.79±0.03 |
| ReACGAN | 4.49±0.10 | 9.84±0.00 |
| PD-GAN | 4.60±0.05 | 9.87±0.06 |
| SONA | **4.24**±0.07 | **10.05**±0.03 |
| *StyleGAN2 backbone* | | |
| ReACGAN | **3.39**±0.03 | 10.33±0.03 |
| PD-GAN | 4.06±0.19 | 10.09±0.05 |
| SONA | 3.38±0.14 | **10.45**±0.08 |

Table 3: TinyImageNet.

| Method | FID ↓ | IS ↑ | Dens ↑ | Cover ↑ | iFID ↓ |
|---|---|---|---|---|---|
| ContraGAN | 23.66±1.59 | 12.47±0.45 | 0.62±0.05 | 0.46±0.03 | 162.69±2.69 |
| ReACGAN | 18.99±0.98 | 15.37±0.68 | 0.70±0.01 | 0.54±0.02 | 130.77±1.22 |
| PD-GAN | 20.77±1.53 | 14.29±1.11 | 0.70±0.05 | 0.58±0.02 | 111.07±3.43 |
| SONA | **16.33**±0.62 | **16.60**±0.35 | **0.74**±0.02 | **0.59**±0.01 | 108.75±0.60 |
| *Apply DiffAug* | | | | | |
| ContraGAN | 11.86±0.32 | 16.01±0.29 | 0.78±0.02 | 0.63±0.01 | 142.07±1.02 |
| ReACGAN | 9.93±0.34 | 20.25±0.07 | 0.88±0.01 | 0.69±0.00 | 107.31±1.22 |
| PD-GAN | 13.09±1.00 | 16.57±0.34 | 0.78±0.02 | 0.70±0.02 | 95.62±2.27 |
| SONA | **7.76**±0.29 | **23.00**±0.10 | **0.99**±0.01 | **0.79**±0.00 | **82.23**±0.48 |

For evaluation, we use Frechét Inception Distance (FID) (Heusel et al., 2017), Inception Score (IS) (Salimans et al., 2016), Density & Coverage (Naeem et al., 2020), intra FID (Miyato & Koyama, 2018, iFID) that is the average of class-wise FID.

We first train SONA on CIFAR10, and report the results in Table 2. We evaluate both Big-GAN (Brock et al., 2019) and StyleGAN2 (Karras et al., 2020b) backbones. The results show that SONA consistently achieves the best performance across all metrics.

Next, we scale up the empirical evaluation by increasing both the image resolution (64×64) and the number of classes (200) using TinyImageNet. As shown in Table 3, SONA outperforms the other SoTA models on all metrics. Notably, SONA also benefits from DiffAug (Zhao et al., 2020), a leading data augmentation technique, achieving the best overall scores.

Finally, we evaluate SONA on the ImageNet dataset at a resolution of 128×128. We use the BigGAN backbone, as it is the only architecture among single-stage generation pipelines capable of producing reasonable images on the dataset. We compare performance under two batch size settings (256 and 2048), with results summarized in Table 4. According to the table, SONA outperforms other methods on all

Table 4: ImageNet.

| Method | FID ↓ | IS ↑ | Dens ↑ | Cover ↑ | iFID ↓ | Top-1/5 acc↑ |
|---|---|---|---|---|---|---|
| *Batch size = 256* | | | | | | |
| ContraGAN | 31.73 | 23.93 | 0.57 | 0.28 | 169.65 | 0.02 / 0.09 |
| ReACGAN | 18.73 | 51.29 | **0.85** | 0.46 | 131.83 | 0.20 / 0.48 |
| PD-GAN | 29.76 | 27.17 | 0.45 | 0.35 | 119.07 | 0.24 / 0.48 |
| SONA | **13.17** | **83.33** | 0.79 | **0.59** | **74.33** | **0.62 / 0.87** |
| *Batch size = 2048* | | | | | | |
| ReACGAN | 8.44 | 103.07 | **1.04** | 0.71 | 87.77 | 0.51 / 0.82 |
| PD-GAN | 8.85 | 96.11 | 0.95 | 0.81 | 52.65 | 0.63 / 0.83 |
| SONA | **6.14** | **140.14** | 1.03 | **0.82** | **48.45** | **0.80 / 0.93** |

metrics except Density. Additionally, we compute Top-1 and Top-5 classification accuracies for the 1,000 ImageNet classes using an Inception V3 network, following Kang et al. (2023a). The results indicate that images generated by SONA align best with the conditioning class among all baselines.

Table 5: Text-to-image generation tasks on CUB and COCO.

| Method | CUB (Wah et al., 2011) | | COCO (Lin et al., 2014) | |
|---|---|---|---|---|
| | FID ↓ | CLIP Score ↑ | FID ↓ | CLIP Score ↑ |
| GALIP (original; concat) | 11.76 | 0.3310 | 5.30 | 0.3639 |
| GALIP + SONA | **10.20** | **0.3342** | **4.70** | **0.3677** |

Table 6: Text-to-image generation task under zero-shot setting on COCO. GALIP and GALIP + SONA are trained on CC12M, while the other baselines are trained on larger-scale datasets.

| Method | Type | Param size (B) | Data size (M) | zFID$_{30K}$ ↓ | CLIP Score ↑ | Speed (sec) |
|---|---|---|---|---|---|---|
| LDM (Rombach et al., 2022) | Diffusion | 1.45 | 400 | 12.63 | - | 3.7 |
| GLIDE (Nichol et al., 2022) | Diffusion | 5 | 250 | 12.24 | - | 15.0 |
| DALL·E 2 (Ramesh et al., 2022) | Diffusion | 6.5 | 250 | 10.39 | - | - |
| Imagen (Saharia et al., 2022) | Diffusion | 7.9 | 860 | 7.27 | - | 9.1 |
| InstaFlow (Liu et al., 2024) | Flow | 0.9 | - | 13.10 | - | 0.09 |
| StyleGAN-T (Sauer et al., 2023) | GAN | 1.02 | 250 | 13.9 | - | 0.10 |
| GALIP (original; concat) | GAN | 0.24+0.08 | 12 | 13.78 | 0.3306 | 0.04 |
| GALIP + SONA | GAN | 0.24+0.08 | 12 | **12.43** | **0.3411** | 0.04 |

## 6.2 BENCHMARK ON TEXT-CONDITIONAL GENERATION TASKS

We demonstrate the applicability of SONA to text-to-image generation tasks. Our experiments are based on GALIP (Tao et al., 2023), which we verified to be reproducible using the official repository[3]. The GALIP discriminator consists of frozen pre-trained CLIP encoders and learnable mod-

---

[3]https://github.com/tobran/GALIP

ules. Text conditioning is performed by concatenating image features and text embeddings from the CLIP encoder, followed by processing with a shallow network. We apply SONA to the discriminator in a straightforward manner, using the frozen CLIP text embedding for $\omega_y$ without modification. To assess the effectiveness of the proposed method, we train both the original GALIP and the SONA-based GALIP on CUB (Wah et al., 2011), COCO (Lin et al., 2014), and CC12M (Changpinyo et al., 2021), respectively. For models trained on CC12M, we report zero-shot performances on COCO. As shown in Table 5 and Table 6, SONA achieves improved FID scores while maintaining comparable text alignment to the original GALIP on three standard image datasets at 256×256 resolution. We suspect that our method reduces interference between the assessment of naturalness and alignment, even with fixed $\omega_y$. Adopting learnable $\omega_y$ for further improvement in CLIP score is left for future work. For reference, we also include other text-to-image models, not limited to GANs, in Table 6. Although these models differ in dataset scale and a direct comparison is not strictly fair because GALIP is trained on the smallest dataset, the table indicates that SONA applied to GALIP achieves competitive generation performance with the fastest inference speed among the listed models.

### 6.3 ABLATION STUDY

We evaluate the contribution of each proposed component in SONA by training models on CIFAR10 using the PyTorch official codebase[4] provided by Brock et al. (2019). Results are summarized in Table 7.

Table 7: Ablation study using CIFAR10

| Adaptive weighting $s$ | Orthogonal proj. in Eq. (6) | matching loss $\mathcal{V}_{\text{BT-M}}$ | FID $\downarrow$ | IS $\uparrow$ |
|---|---|---|---|---|
| ✓ | | | 7.51±0.14 | 9.08±0.07 |
| ✓ | ✓ | | 6.29±0.08 | 9.14±0.04 |
| ✓ | | ✓ | 6.02±0.28 | 9.54±0.82 |
| ✓ | ✓ | ✓ | 5.65±0.25 | 9.51±0.05 |
| | ✓ | ✓ | 7.09±1.17 | 9.52±0.07 |

ble 7. Orthogonal modeling in $f_{\Phi_A}^A(x, y)$ improves the generation performance in FID, while the BT-M loss $\mathcal{V}_{\text{BT-M}}$ does in IS. By adopting both, SONA achieves better generation performance in terms of both FID and IS. In contrast, we can also see that the adaptive scaling coefficients introduced in Section 4.4 work.

### 6.4 DISCUSSION ON COMPUTATIONAL TIME

We report the computational efficiency of each baseline in Table 8, based on the experiments described in Section 6.1. PD-GAN demonstrates the highest training efficiency, attributable to its simple design. Nevertheless, SONA achieves comparable efficiency to PD-GAN and surpasses other state-of-the-art classifier-based methods. Additionally, we illustrate the training convergence behavior on ImageNet (with a batch size of 2048) in Section 6.1, as shown in Figure 4. We observe that SONA attains FID scores similar to PD-GAN for approximately the first three days, after which SONA exhibits a clear improvement, achieving lower FID scores. This performance gain justifies the additional computational overhead of SONA compared to PD-GAN.

Table 8: Training efficiency (iteration/min).

| Method | CIFAR10 | TinyIN | ImageNet |
|---|---|---|---|
| ContraGAN | 360.36 | 129.87 | 80.70 |
| ReACGAN | 322.15 | 107.23 | 78.84 |
| PD-GAN | **442.80** | **195.76** | **101.91** |
| SONA | 410.95 | 169.13 | 90.16 |

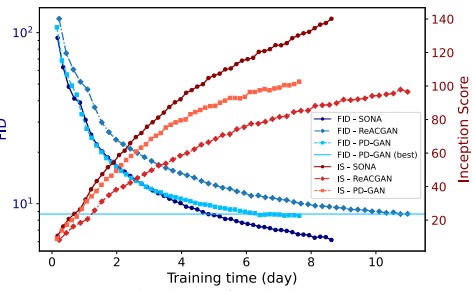

Figure 4: Behavior of training convergence: FID ($\downarrow$) and IS ($\uparrow$) as a function of training time (in days).

### 7 CONCLUSION

In this paper, we introduced SONA, a novel discriminator framework for conditional GANs that efficiently evaluates both sample naturalness (authenticity) and conditional alignment, while adaptively balancing unconditional, conditional, and matching-aware discrimination objectives. Experiments on image datasets demonstrate that generators trained with our method produce higher-quality samples that are more accurately aligned with the given labels compared to state-of-the-art methods. Additionally, we showed that SONA is applicable to text-to-image generation scenarios.

---

[4]https://github.com/ajbrock/BigGAN-PyTorch

## ETHICS STATEMENT

Because our work involves training AI models that can generate synthetic content, there are inherent risks of producing harmful or inappropriate outputs, such as deepfake images, graphic violence, offensive material, or content that may infringe on copyright. To mitigate these risks, it is essential to implement robust content filtering and moderation measures to prevent the creation of unethical, harmful, or infringing media.

## REPRODUCIBILITY STATEMENT

All experiments described in Section 6 were implemented using open-source repositories, which we confirm are reproducible by rerunning them. The datasets employed in this study are publicly available via their official sources. Detailed implementation procedures are provided in Appendix F. Additionally, we provide codes as supplementary material and outline our training procedure in Algorithm 1. The proofs of our theoretical claims can be found in Appendix.

## LLM USAGE

Large Language Models (LLMs) were used for academic proofreading and assistance in writing the abstract. They also supported coding tasks, including debugging, resolving errors, and visualizing results. All research ideas and theoretical contributions were developed solely by the authors.

## ACKNOWLEDGMENTS

We are grateful to Yuya Kobayashi for assistance in ensuring the reproducibility of the GALIP code and to Masato Ishii for helpful comments during the preparation of this manuscript. We also acknowledge the anonymous reviewers for their valuable suggestions and comments.

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

# A    RELATED WORKS

The first (class-)conditional GAN was introduced by Mirza & Osindero (2014), who incorporated class information by concatenating the input with the corresponding class embedding. This straightforward approach has been widely adopted in subsequent works (Reed et al., 2016; Zhang et al., 2017; Tao et al., 2023; Kang et al., 2023b). For conditional discriminators, it has been shown to be more effective to concatenate class information with intermediate discriminator features rather than directly with the input (Reed et al., 2016), a strategy now used in several modern text-to-image GANs (Tao et al., 2023; Kang et al., 2023b).

The projection-based approach, introduced by Miyato & Koyama (PD-GAN; 2018), has proven effective for both generation quality and conditional alignment, despite its simplicity. Like the concatenation-based approach, it requires only minor modifications to the discriminator's final projection layer and no further architectural changes, facilitating scalability and extensibility. While the concatenation method is similar to the projection-based approach—especially when using the deepest discriminator features—the projection-based method has been empirically shown to be more effective in class-conditional settings due to its well-designed inductive bias based on probabilistic modeling. This approach is now widely used in conditional generation tasks (Brock et al., 2019; Karras et al., 2019; 2020b; 2021; Sauer et al., 2022; 2023; Huang et al., 2024) and has been extended to more challenging scenarios, such as text-to-image generation (Sauer et al., 2023), where the set of possible text prompts is not finite.

As a more explicit approach to enforcing conditional alignment, Odena et al. (2017) proposed the auxiliary classifier GAN (AC-GAN), which adds a classifier to the discriminator to predict class labels of generated images. AC-GAN combines the standard GAN loss with a cross-entropy classification loss. However, AC-GANs have been observed to suffer from limited diversity in generated samples (Shu et al., 2017), a limitation attributed to the absence of a negative conditional entropy term in the objective (Shu et al., 2017; Gong et al., 2019). Later works addressed this by applying the classification loss to both real and generated samples (Gong et al., 2019; Hou et al., 2022). Kang et al. (2021) identified instability in AC-GAN training due to unbounded discriminator features and poor early-stage classification, and proposed ReACGAN to address these issues. Separately, Kang & Park (2020) introduced ContraGAN, which incorporates data-to-data relations in addition to data-to-class relations (Equation (4)).

As shown in our experiments (Section 6.1) and recent benchmarks (Kang et al., 2023a), ReACGAN achieves SoTA performance on widely used class-image datasets among conditional discriminator methods, including projection-based approaches. However, to our knowledge, this approach has not been extended beyond class-conditional settings, such as text-to-image tasks, likely due to greater implementation complexity and higher computational cost compared to projection-based and concatenation-based methods. Moreover, extending $Y$ beyond a finite discrete set (e.g., to text prompts) in this approach is generally non-trivial. Specifically, the classification loss adopted in this approach is equivalent to the InfoNCE loss (Equation (4)), which considers all plausible negative labels $y' \sim p_{\rm d}(y')$ for each training sample $y \sim p_{\rm d}(y|x)$. Therefore, extending $Y$ beyond a finite discrete set (e.g., to text prompts) in this approach is generally non-trivial. In contrast, while BT-C and BT-M losses in SONA can also be interpreted as variants of the InfoNCE loss, as discussed around Proposition 4, they use only a single negative sample $x'$ drawn from $p_{\rm d}(x')$ per training sample $x$ drawn from $p_{\rm d}(x|y)$. This has two main advantages: First, this loss is easily applicable to general conditioning cases, including text-to-image, because sampling a single negative sample is feasible. Second, this property of our losses greatly reduces computational complexity, as reported in Table 8.

For text-to-image GANs, which are more challenging than class-conditional generation tasks, both concatenation-based and projection-based approaches have recently been adopted. Kang et al. (2023b) and Tao et al. (2023) employed the concatenation-based approach, injecting frozen CLIP-encoded text embeddings into deep discriminator features. In contrast, Sauer et al. (2023) adopted the projection-based approach, modeling text-conditional projections by applying a learnable affine transformation to frozen CLIP text embeddings. In addition to discriminator design, these works introduced additional losses to improve text-conditional alignment. Notably, a matching loss uses negative pairs of images and text prompts as fake samples (Kang et al., 2023b; Tao et al., 2023), analogous to our loss $\mathcal{V}_{\rm BT\text{-}M}$. Furthermore, all three works employed a CLIP-guidance loss, which maximizes the cosine similarity between CLIP embeddings of the text condition and generated images. A

similar technique using an ImageNet classifier was applied to class-conditional GANs (Sauer et al., 2022).

## B  SUPPLEMENT FOR PRELIMINARY CONCEPTS

We review three key concepts, GAN, SAN, and Bradly–Terry model, which are background of our method.

### B.1  GAN

**GAN.** Goodfellow et al. (2014) originally formulated GANs as a two-player game between a generator and a discriminator. The generator aims to produce realistic samples that can fool the discriminator, while the discriminator seeks to distinguish real samples from the data distribution and fake samples generated by the generator, outputting a scalar value. Based on this framework, two variants of GAN minimax objectives were proposed. The first, known as the saturating GAN objective, is defined as:

$$\mathcal{V}_{\text{ORIG-GAN}}(f) = \mathbb{E}_{p_{\text{data}}(x)}[\log(\sigma(f(x)))] + \mathbb{E}_{p_g(x)}[\log(1 - \sigma(f(x)))] \tag{14}$$

$$\mathcal{J}_{\text{S-GAN}}(g) = \mathbb{E}_{p_g(x)}[\log(1 - \sigma(f(x)))], \tag{15}$$

The second variant, referred to as the non-saturating GAN objective, shares the same maximization objective but uses a different minimization objective:

$$\mathcal{J}_{\text{NS-GAN}}(g) = -\mathbb{E}_{p_g(x)}[\log(\sigma(f(x)))] \tag{16}$$

It is well established that the global minimum of $\mathcal{J}_{\text{S}}$ and $\mathcal{J}_{\text{NS}}$, when $f$ maximizes $\mathcal{V}_{\text{ORIG}}$, is achieved if and only if $p_g = p_{\text{data}}$.

The maximization objective can be equivalently rewritten as:

$$\mathcal{V}_{\text{ORIG-GAN}}(f) = \mathbb{E}_{p_{\text{data}}(x)}[\log(\sigma(f(x)))] + \mathbb{E}_{p_g(x)}[\log(1 - \sigma(f(x)))] \tag{17}$$

$$= \mathbb{E}_{p_{\text{data}}(x)}[\log(\sigma(f(x)))] + \mathbb{E}_{p_g(x)}[\log(\sigma(-f(x)))], \tag{18}$$

which consists solely of $\log \sigma(\cdot)$ terms. We use this maximization objective for $\mathcal{V}_{\text{SAN}}$, which is applied to our unconditional discrimination.

**Relativistic GAN.** Jolicoeur-Martineau (2018) introduced a relativistic variant of GANs, also formulated as a minimax problem but based on a relativistic discriminator. The original relativistic GAN, now known as relativistic pairing GAN (RpGAN), is defined using LogSigmoid as:

$$\mathcal{V}_{\text{LS-RPGAN}}(f) = \mathbb{E}_{p_{\text{data}}(x_{\text{d}})p_g(x_g)}[\log \sigma(f(x_{\text{d}}) - f(x_g))] \tag{19}$$

$$\mathcal{J}_{\text{LS-RPGAN}}(g) = -\mathbb{E}_{p_{\text{data}}(x_{\text{d}})p_g(x_g)}[\log \sigma(f(x_g) - f(x_{\text{d}}))]. \tag{20}$$

Our BT-C loss, $\mathcal{V}_{\text{BT-C}}$, can be interpreted as a conditional counterpart to Equation (19). Accordingly, we define our minimization loss for conditional alignment, $\mathcal{J}_{\text{BT-C}}$, as the conditional counterpart to Equation (20).

### B.2  FROM SLICED WASSERSTEIN TO SAN

**Sliced Wasserstein.** Sliced Wasserstein (SW) was introduced as a variant of the Wasserstein distance and has been further developed, resulting in important extensions such as generalized SW (GSW) (Kolouri et al., 2019) and augmented SW (ASW) (Chen et al., 2022). SW leverages a key property of the Wasserstein distance: it admits a closed-form solution when the data space is one-dimensional. The closed-form expression for the Wasserstein distance between one-dimensional distributions with measures $\mu$ and $\nu$ is given by

$$\mathbf{W}_p(\mu, \nu) = \left( \int_0^1 |F_\mu^{-1}(\rho) - F_\nu^{-1}(\rho)| \right), \tag{21}$$

where $F_\mu^{-1}(\cdot)$ denotes the quantile function of the probability measure $\mu$. The main idea of SW is to exploit this closed-form by projecting higher-dimensional probability distributions onto one-dimensional spaces using the Radon transform over a set of directions, defined as

$$\mathcal{R}I(\xi, \omega) = \int I(x)\delta(\xi - \langle x, \omega \rangle)dx, \tag{22}$$

where the higher-dimensional space is projected onto a one-dimensional space with direction $\omega \in \mathbb{S}^{D-1}$. Specifically, SW is defined as

$$\mathrm{SW}_p(\mu, \nu) = \left( \int_{\omega \in \mathbb{S}^{D-1}} \mathrm{W}_p^p(\mathcal{R}I_\mu(\cdot, \omega), \mathcal{R}I_\nu(\cdot, \omega)) d\omega \right)^{1/p}. \tag{23}$$

This decomposition of the higher-dimensional space into a collection of one-dimensional spaces makes SW much more computationally tractable than the original Wasserstein distance.

**Variants of SW.** Kolouri et al. (2019) introduced the generalized sliced Wasserstein (GSW) distance by extending the standard Radon transform to the generalized Radon transform (GRT), defined as

$$\mathcal{G}I(\xi, \omega) = \int I(x)\delta(\xi - g(x, \omega))dx, \tag{24}$$

where $g$ is a defining function that satisfies certain conditions (**H1**-**H4** in Kolouri et al. (2019)). The GRT includes the standard Radon transform as a special case. By replacing the Radon transform in the definition of SW with the GRT, the GSW is formulated as

$$\mathrm{GSW}_p(\mu, \nu) = \left( \int_{\omega \in \mathbb{S}^{D-1}} \mathrm{W}_p^p(\mathcal{G}I_\mu(\cdot, \omega), \mathcal{G}I_\nu(\cdot, \omega)) d\omega \right)^{1/p}. \tag{25}$$

This formulation is simple and enables a broad class of transformations for projecting data samples onto one-dimensional spaces. Building on this generalization, Chen et al. (2022) proposed the augmented sliced Wasserstein (ASW) distance. The ASW is also based on an extension of the Radon transform, specifically the spatial Radon transform, which is defined as

$$\mathcal{S}^h I(\xi, \omega) = \int I(x)\delta(\xi - \langle \omega, h(x) \rangle)dx, \tag{26}$$

where $h$ is any injective function. The ASW is then defined as

$$\mathrm{ASW}_p(\mu, \nu) = \left( \int_{\omega \in \mathbb{S}^{D-1}} \mathrm{W}_p^p(\mathcal{S}^h I_\mu(\cdot, \omega), \mathcal{S}^h I_\nu(\cdot, \omega)) d\omega \right)^{1/p}. \tag{27}$$

Although ASW is a valid distance for distributions, it, as well as SW and GSW, requires dense sampling of $\omega$ on the high-dimensional hypersphere for accurate approximation. To address this computational complexity, the maximum sliced Wasserstein (max-SW) distance was proposed. For example, max-ASW is defined by selecting a single direction that best distinguishes the target probability distributions in the projected one-dimensional space, rather than integrating over all possible directions:

$$\mathrm{max\text{-}ASW}_p^h(\mu, \nu) = \max_{\omega \in \mathbb{S}^{D-1}} \mathrm{W}_p(\mathcal{S}^h I_\mu(\cdot, \omega), \mathcal{S}^h I_\nu(\cdot, \omega)). \tag{28}$$

ASW is guaranteed to be a distance as long as the function $h$ is injective.

**SAN.** Takida et al. (2024) formulated SANs by modifying the discriminator optimization in both the architecture and objective function. The core idea is to design the discriminator to approximately evaluate the max-ASW with only minor modifications. To bridge the gap between GAN optimization and max-ASW evaluation, they propose imposing three key conditions on the discriminator $f(x) = \langle \omega, h(x) \rangle$: (i) direction optimality for $\omega$, (ii) injectivity for $h$, and (iii) separability for $h$. Direction optimality is motivated by the selection of a single direction $\omega$ in Equation (28), while injectivity is necessary to ensure that the discriminator defines a valid distance. A detailed explanation of the third condition, separability, is provided in Appendix E.1.

### B.3 BRADLEY–TERRY FRAMEWORK

The Bradley–Terry framework (Bradley & Terry, 1952) provides a general method for assigning scores to a set of items based on pairwise comparisons. Since its introduction, it has been widely applied to various machine learning problems, particularly in reward modeling using human preference annotations.

The core idea is to model the log-odds that item $x_w$ is preferred over $x_\ell$ as the difference between their scores. Specifically, the preference scoring function, denoted as $\tilde{f}$, is learned to represent the log-odds difference between items $x_w$ and $x_\ell$ as follows:

$$\Pr(x_w \text{ is preferred over } x_\ell | y) = \sigma(\tilde{f}(x_w, y) - \tilde{f}(x_\ell, y)). \tag{29}$$

Hereafter, we refer to $x_w$ and $x_\ell$ as the winning and losing samples, following the convention in reinforcement learning from human feedback (RLHF). The function $\tilde{f}$ is learned by maximizing the following objective:

$$\mathcal{V}_{\text{BT}} = \mathbb{E}_{x_w, x_\ell}[\log \sigma(\tilde{f}(x_w) - \tilde{f}(x_\ell))], \tag{30}$$

When preferences are conditioned on additional information $y$, the objective becomes

$$\mathcal{V}_{\text{BT}} = \mathbb{E}_{x_w, x_\ell, y}[\log \sigma(\tilde{f}(x_w, y) - \tilde{f}(x_\ell, y))], \tag{31}$$

which is equivalent to Equation (8).

In our setup, we use the discriminator $f$ as the scoring function. In typical problem setups, a dataset of paired samples with preference labels is available. However, in our case, such a dataset is not provided in the required format. Instead, we construct pairs of winning and losing samples in two distinct ways under reasonable assumptions. First, we assume that a sample randomly selected from the dataset is always preferred over a generated sample. Under this assumption, the joint distribution of winning and losing samples is defined as $p(x_w, x_\ell, y) = p_{\text{data}}(y) p_{\text{data}}(x_w | y) p_g(x_\ell | y)$, resulting in $\mathcal{V}_{\text{BT-C}}$. Second, we assume that a sample from a subset of the dataset associated with a given condition is always preferred over a data sample selected without regard to the condition. In this case, the joint distribution is represented as $p(x_w, x_\ell, y) = p_{\text{data}}(y) p_{\text{data}}(x_w | y) p_{\text{data}}(x_\ell)$, resulting in $\mathcal{V}_{\text{BT-M}}$.

## C  Algorithm

Please refer to Algorithm 1 for the pseudo code describing GAN training with SONA. Note that, in the application to GALIP (Section 6.2), we use frozen CLIP text embeddings to model $\omega_y$, which does not involve the optimization of $\omega$ in Algorithm 1.

---

**Algorithm 1** GAN Training with SONA

---

**Input:** Data distribution $p_{\text{data}}$; latent distribution $p_Z$; generator parameters $\theta$; discriminator parameters $\Phi = (\omega, \omega_y, \psi)$, where $\psi$ models $h$ as $h_\psi$; parameters for learnable weighting $(\tilde{s}_{\text{SAN}}, \tilde{s}_{\text{BT-C}}, \tilde{s}_{\text{BT-M}})$; batch size $N$; learning rates $(\eta_\theta, \eta_\omega, \eta_{\omega_y}, \eta_\psi, \eta_{\tilde{s}})$; total iterations $T$; update ratio $I$.

**for** $t = 1, 2, \ldots, T$ **do**
    **for** $i = 1, 2, \ldots, I$ **do**
        Obtain weight coefficient sets by
            $(s_{\text{SAN}}, s_{\text{BT-C}}, s_{\text{BT-M}}) = \texttt{Normalize} \circ \texttt{Softplus}(\tilde{s}_{\text{SAN}}, \tilde{s}_{\text{BT-C}}, \tilde{s}_{\text{BT-M}})$
        Sample minibatch $\{(x_{\text{data},n}, y_n)\}_{n \in [N]}$ from $p_{\text{data}}$
        Sample latent variables $\{z_n\}_{n \in [N]}$ from $p_Z$
        Generate synthetic samples $x_{\text{gen},n} = g_\theta(z_n, y_n)$ for $n \in [N]$
        Create negative samples $x_{\text{neg},n} = x_{\text{data},\pi(n)}$ using a random permutation $\pi$
        Compute $\mathcal{V}_{\text{SAN}}$, $\mathcal{V}_{\text{BT-C}}$, and $\mathcal{V}_{\text{BT-M}}$ with $\{(x_{\text{data},n}, x_{\text{neg},n}, x_{\text{gen},n}, y_n)\}_{n \in [N]}$
        Update $\omega \leftarrow \omega + \eta_\omega \nabla_\omega(\mathcal{V}_{\text{SAN}} + \mathcal{V}_{\text{BT-C}} + \mathcal{V}_{\text{BT-M}})$
        Update $\psi \leftarrow \psi + \eta_\psi \nabla_\psi(\mathcal{V}_{\text{SAN}} + \mathcal{V}_{\text{BT-C}} + \mathcal{V}_{\text{BT-M}})$
        Update $\tilde{s} \leftarrow \tilde{s} + \eta_\psi \nabla_{\tilde{s}}(\mathcal{V}_{\text{SAN}} + \mathcal{V}_{\text{BT-C}} + \mathcal{V}_{\text{BT-M}})$
        Update $\omega_y \leftarrow \omega_y + \eta_{\omega_y} \nabla_{\omega_y}(\mathcal{V}_{\text{BT-C}} + \mathcal{V}_{\text{BT-M}})$
    **end for**
    Sample minibatch $\{(x_{\text{data},n}, y_n)\}_{n \in [N]}$ from $p_{\text{data}}$
    Sample latent variables $\{z_n\}_{n \in [N]}$ from $p_Z(z)$
    Generate synthetic samples $x_{\text{gen},n} = g_\theta(z_n, y_n)$ for $n \in [N]$
    Compute $\mathcal{J}_{\text{SAN}}$ and $\mathcal{J}_{\text{BT-C}}$ with $\{(x_{\text{data},n}, x_{\text{gen},n}, y_n)\}_{n \in [N]}$
    Update $\theta \leftarrow \theta - \eta_\theta \nabla_\theta(\mathcal{J}_{\text{SAN}} + \mathcal{V}_{\text{BT-C}})$
**end for**

---

## D  Analysis of Existing Approaches

### D.1  Proposition 1

We introduce the following lemma, which is taken from the proof of Zhang et al. (2023, Proposition 1).

**Lemma 5.** *The function $\tilde{f}$ maximizes $\mathcal{V}_{\text{CE}}$ if $\tilde{f}(x, y) = \log p_{\text{data}}(x|y) + r_X(x)$ for an arbitrary function $r_X : X \to \mathbb{R}$.*

**Proposition 1** (Log conditional probability maximizes $\mathcal{V}_{\text{CE}}$). *Assume $p_{\text{data}}(y)$ is a constant regardless of $y \in Y$, e.g., a uniform distribution. The function $\tilde{f}$ maximizes $\mathcal{V}_{\text{CE}}$ if $\tilde{f}(x, y) = \log p_{\text{data}}(y|x) + r_X(x)$ for an arbitrary function $r_X : X \to \mathbb{R}$.*

*Proof.* By Lemma 5, the maximizer $\tilde{f}$ can be written as

$$\tilde{f}(x, y) = \log p_{\text{data}}(x|y) + r'_X(x), \tag{32}$$

where $r'_X : X \to \mathbb{R}$ is an arbitrary function. By Bayes' theorem, we have

$$\log p_{\text{data}}(x|y) = \log p_{\text{data}}(y|x) + \log p_{\text{data}}(x) - \log p_{\text{data}}(y) \tag{33}$$
$$= \log p_{\text{data}}(y|x) - \log p_{\text{data}}(x) + C, \tag{34}$$

where $C$ denotes the constant $\log p_{\text{data}}(y)$, since $p_{\text{data}}(y)$ is assumed to be constant. Substituting Equation (34) into Equation (32) completes the proof. $\qquad\square$

# E    ANALYSIS OF SONA

## E.1    FORMAL STATEMENT OF PROPOSITION 2

We formally state Proposition 2 in this section.

First, we introduce two key assumptions required for this proposition. To do so, we present the concept of separability from Takida et al. (2024), which is used to formulate assumptions on the function $h$ in the discriminator. This property is important for ensuring that the discriminator induces a meaningful distance between target distributions.

The definition of separability relies on the spatial Radon Transform (Chen et al., 2022, SRT), defined as follows:

**Definition 1.** (Spatial Radon Transform) Given a measurable injective function $h : X \to \mathbb{R}^D$ and a function $U : X \to \mathbb{R}$, the spatial Radon transform of $U$ is

$$\mathcal{S}^h U(\cdot, \omega) = \int_X U(x) \delta(\xi - \langle \omega, h(x) \rangle) dx, \tag{35}$$

where $\xi \in \mathbb{R}$ and $\omega \in \mathbb{S}^{D-1}$ parameterize the hypersurfaces $\{x \in X \mid \langle \omega, h(x) \rangle = \xi\}$.

The SRT generalizes the Radon Transform using an injective function. If $U$ is a probability density, the SRT corresponds to applying the standard Radon transform to the pushforward of $U$ by $h$. In this case, intuitively, the SRT projects $h(x)$ onto a scalar along direction $\omega$ with the probability. One of its crucial properties is that, for two probability densities $p$ and $q$, if $\mathcal{S}^h p(\xi, \omega) = \mathcal{S}^h q(\xi, \omega)$ is satisfied for all $\xi \in \mathbb{R}$ and $\omega \in \mathbb{S}^{D-1}$, then $p = q$ holds due to the injectivity of $h$. Thus, an injective $h$ preserves information about the equality of target distributions. This leads to our first assumption:

**Assumption A.** *We assume that $h : X \to \mathbb{R}^D$ is injective.*

Using the SRT, we define separability as follows:

**Definition 2.** (Separable) Given probability densities $p$ and $q$ on $X$, and $h : X \to \mathbb{R}^D$, let $\omega \in \mathbb{S}^{D-1}$, and let $F_p^{h,\omega}(\cdot)$ denote the cumulative distribution function of $\mathcal{S}^h p(\cdot, \omega)$. If $\omega^* = \arg\max_\omega \mathbb{E}_{p(x)}[\langle \omega, h(x) \rangle] - \mathbb{E}_{q(x)}[\langle \omega, h(x) \rangle]$ satisfies $F_p^{h,\omega^*}(\xi) \leq F_q^{h,\omega^*}(\xi)$ for all $\xi \in \mathbb{R}$, then $h$ is **separable** for $p$ and $q$.

Intuitively, separability ensures that the optimal transport map in the one-dimensional space induced by the SRT from $\mathcal{S}^h p(\cdot, \omega^*)$ to $\mathcal{S}^h q(\cdot, \omega^*)$ is aligned in the same direction for all samples. This suggests that $h$ can bring $p$ and $q$ closer, at least along the optimal direction $\omega^*$, which also maximizes $\mathcal{V}_{\text{SAN}}(\omega, h)$ for a given $h$. Thus, we make our second assumption:

**Assumption B.** *We assume that $h : X \to \mathbb{R}^D$ is separable for $p_{\text{data}}(x)$ and $p_g(x)$.*

With these assumptions, we can now formally state Proposition 2.

**Proposition 2** (Formal; Unconditional discrimination by $\mathcal{V}_{\text{SAN}}$)**.** *Let the discriminator be $f(x) = \langle \omega, h(x) \rangle$ with $\omega \in \mathbb{R}^{D-1}$ and $h : X \to \mathbb{R}^D$. Suppose Assumptions A and B hold, and let $\hat{\omega} = \arg\max_\omega \mathcal{V}_{\text{SAN}}(\omega, h)$ for a given $h$. Then, the objective $\mathcal{J}_{\text{SAN}}(g; \hat{\omega}, h)$ is minimized if and only if $g$ minimizes the following functional mean divergence between $p_{\text{data}}(x)$ and $p_g(x)$, given by*

$$\text{FM}^*(p_{\text{data}}, p_g) = \left\| \mathbb{E}_{p_{\text{data}}(x)}[h(x)] - \mathbb{E}_{p_g(x)}[h(x)] \right\|_2, \tag{36}$$

*which is a valid distance under these assumptions.*

The proof of Proposition 2 is provided in Takida et al. (2024).

## E.2    PROPOSITION 3

We introduce a lemma, which is a restatement of a portion of claims made in Theorem 3.1 of Jolicoeur-Martineau (2020).

**Lemma 6.** *Let $v : \mathbb{R} \to \mathbb{R}$ be a concave function such that $v(0) = C$, $v$ is differentiable at $0$, $v'(0) \neq 0$, $\sup_t(v(t)) > 0$, and $\arg\sup_t(v(t)) > 0$. Let $p$ and $q$ be probability distributions with support $X$. Then, $\sup_f \mathbb{E}_{p(x)q(x')}[v(f(x) - f(x'))]$ is a divergence, up to $C$.*

**Proposition 3** (Conditional learning by $\mathcal{V}_{\text{BT-C}}$). *Let the discriminator be $f(x, y) = f^{\text{N}}(x) + f^{\text{A}}(x, y)$, where $f^{\text{N}}(x) = \langle \omega, h(x) \rangle$ with $\omega \in \mathbb{S}^{D-1}$, $h : X \to \mathbb{R}^D$, $b \in \mathbb{R}$, and $f^{\text{A}} : X \times Y \to \mathbb{R}$. Assume that the generator achieves $p_g(x) = p_{\text{data}}(x)$, and $\omega$ and $h$ maximize $\mathcal{V}_{\text{SAN}}$ for given $p_{\text{data}}$ and $p_g$. If $f^{\text{A}}$ maximizes Equation* (13), *then it is minimized if and only if $p_{\text{data}}(x|y) = p_g(x|y)$ for $y \in Y$.*

$$\mathcal{L}_{\text{BT-C}} = \mathbb{E}_{p_{\text{data}}(y)p_{\text{data}}(x_w|y)p_g(x_\ell|y)}[\log \sigma(f^{\text{N}}(x_w) + f^{\text{A}}(x_w, y) - f^{\text{N}}(x_\ell) - f^{\text{A}}(x_\ell, y))].$$

*Proof.* Given $p_g(x) = p_{\text{data}}(x)$ and the optimality of $f^{\text{N}}$ for the specified $p_g$ and $p_{\text{data}}$, it follows that $f^{\text{N}}(x) = C$ for all $x \in X$, where $C \in \mathbb{R}$ is a constant. Substituting this into Equation (9), we obtain:

$$\mathcal{L}_{\text{BT-C}}(\omega_y, h) = \mathbb{E}_{p_{\text{data}}(y)p_{\text{data}}(x_w|y)p_g(x_\ell|y)}\left[\log \sigma\big((C + f^{\text{A}}(x_w, y)) - (C + f^{\text{A}}(x_\ell, y))\big)\right]$$

$$= \mathbb{E}_{p_{\text{data}}(y)p_{\text{data}}(x_w|y)p_g(x_\ell|y)}\left[\log \sigma\big(f^{\text{A}}(x_w, y) - f^{\text{A}}(x_\ell, y)\big)\right]. \tag{37}$$

Since $\log \sigma(\cdot)$ satisfies the conditions of Lemma 6, applying this lemma to $\mathcal{V}_{\text{BT-C}}$ and $\mathcal{V}_{\text{BT-C}}$ establishes the claim. $\square$

**Note on the assumption of distribution matching.** In Proposition 3, we assume $p_g(x) = p_{\text{data}}(x)$ for theoretical development, which we acknowledge is a rather strong condition. We empirically verify that the effectiveness of the conditional alignment term in SONA does not depend on this assumption being satisfied in practice. To demonstrate this, we plot the FID score and Top-1 accuracy with respect to training steps for ImageNet (batch size 2048) in Section 6.1. This figure shows that conditional alignment improves from the beginning, even when there is still a deviation between $p_g(x)$ and $p_{\text{data}}(x)$ in terms of FID, as observed in ReACGAN and PD-GAN.

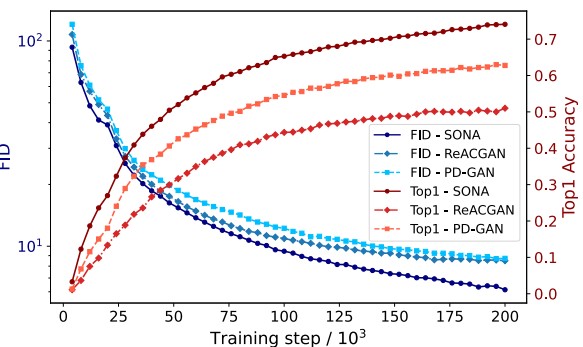

Figure B: FID ($\downarrow$, an indicator of generation performance) and Top-1 accuracy ($\uparrow$, an indicator of conditional alignment) as a function of training iteration.

### E.3    PROPOSITION 4

We introduce a lemma (presented in the proof of Oko et al. (2025, Lemma 1) in the discrete case), which will come in handy for the proof of Proposition 4.

**Lemma 7.** *Consider minimizing $\mathcal{V}_{\text{InfoNCE}}$ over all possible functions $\tilde{f} : X \times Y \to \mathbb{R}$.*

$$\mathcal{V}_{\text{InfoNCE}}(\tilde{f}) = \mathbb{E}_{p_{\text{data}}(x,y)p_{\text{data}}(x')}\left[\log \frac{\exp(\tilde{f}(x, y))}{\exp(\tilde{f}(x, y)) + \exp(\tilde{f}(x', y))}\right]. \tag{38}$$

*$\mathcal{V}_{\text{InfoNCE}}$ is maximized if $\tilde{f}(x, y) = \log p_{\text{data}}(y|x) + r_Y(y)$ for an arbitrary function $r_Y : Y \to \mathbb{R}$.*

*Proof.* This proof is essentially a modification of the proof of Oko et al. (2025, Lemma 1) to our case. Let $q_{x_0, x_1, y}$ and $q^{\tilde{f}}_{x_0, x_1, y}$ be probability mass functions over $\{0, 1\}$ given by

$$q_{x_0, x_1, y}(0) = \frac{p_{\text{data}}(x_0, y)p_{\text{data}}(x_1)}{p_{\text{data}}(x_0, y)p_{\text{data}}(x_1) + p_{\text{data}}(x_1, y)p_{\text{data}}(x_0)} = \frac{p_{\text{data}}(y|x_0)}{p_{\text{data}}(y|x_0) + p_{\text{data}}(y|x_1)}$$

and

$$q^{\tilde{f}}_{x_0, x_1, y}(0) = \frac{\exp(\tilde{f}(x_0, y))}{\exp(\tilde{f}(x_0, y)) + \exp(\tilde{f}(x_1, y))}.$$

Then, we have

$$
\mathcal{V}_{\text{InfoNCE}}(\tilde{f}) = \frac{1}{2}(\mathbb{E}_{p_{\text{data}}(x_0,y)p_{\text{data}}(x_1)}[\log q_{x_0,x_1,y}^{\tilde{f}}(0)] + \mathbb{E}_{p_{\text{data}}(x_1,y)p_{\text{data}}(x_0)}[\log q_{x_0,x_1,y}^{\tilde{f}}(1)])
$$

$$
= \mathbb{E}_{\frac{1}{2}(p_{\text{data}}(x_0,y)p_{\text{data}}(x_1)+p_{\text{data}}(x_1,y)p_{\text{data}}(x_0))}[q_{x_0,x_1,y}(0)\log q_{x_0,x_1,y}^{\tilde{f}}(0) + q_{x_0,x_1,y}(1)\log q_{x_0,x_1,y}^{\tilde{f}}(1)]
$$

$$
= \mathbb{E}_{\frac{1}{2}(p_{\text{data}}(x_0,y)p_{\text{data}}(x_1)+p_{\text{data}}(x_1,y)p_{\text{data}}(x_0))}[-H(q_{x_0,x_1,y}, q_{x_0,x_1,y}^{\tilde{f}})],
$$

where $H(q, q^{\tilde{f}}) = \mathbb{E}_q[-\log q^{\tilde{f}}]$ is the cross entropy, which is minimized when $q^{\tilde{f}} = q$. Since $q^{\tilde{f}} = q$ holds when $\tilde{f}(x,y) = \log p_{\text{data}}(y|x) + r_Y(y)$ for a function $r_Y$, we have proven the assertion. $\qquad\square$

**Proposition 4** (Log gap probability maximizes $\mathcal{V}_{\text{BT-M}}$). *The function $\tilde{f}$ maximizes $\mathcal{V}_{\text{BT-M}}$ if $\tilde{f}(x,y) = \log p_{\text{data}}(x|y) - \log p_{\text{data}}(x) + r_Y(y)$ for an arbitrary function $r_Y : Y \to \mathbb{R}$.*

*Proof.* The objective $\mathcal{V}_{\text{BT-M}}$ is reformulated as

$$
\mathcal{V}_{\text{BT-M}}(\tilde{f}) = \mathbb{E}_{p_{\text{data}}(y)p_{\text{data}}(x_w|y)p_{\text{data}}(x_\ell)}[\log \sigma(\tilde{f}(x_w,y) - \tilde{f}(x_\ell,y))] \tag{39}
$$

$$
= \mathbb{E}_{p_{\text{data}}(x_w,y)p_{\text{data}}(x_\ell)}\left[\log \frac{\exp(\tilde{f}(x_w,y))}{\exp(\tilde{f}(x_w,y)) + \exp(\tilde{f}(x_\ell,y))}\right] \tag{40}
$$

Equation (40) is now equivalent to $\mathcal{V}_{\text{InfoNCE}}$. Therefore, the claim has been proven as a direct consequence of Lemma 7 and Bayes' theorem: $\log p_{\text{data}}(y|x) = \log p_{\text{data}}(x|y) - \log p_{\text{data}}(x) + \log p_{\text{data}}(y)$. $\qquad\square$

Note that, from the proof of Lemma 7, we can also prove the "only if" statement up to some $p_{\text{data}}$-null sets.

### E.4 INSIGHT INTO ADAPTIVE WEIGHTING

We provide insight into the adaptivity of our proposed weighting scheme, which employs learnable scalar parameters $s_{\text{SAN}}^2$, $s_{\text{BT-C}}^2$, and $s_{\text{BT-M}}^2$, by simplifying the maximization objectives.

Recall that the maximization objectives $\mathcal{V}_{\text{SAN}}$, $\mathcal{V}_{\text{BT-C}}$, and $\mathcal{V}_{\text{BT-M}}$ can be expressed with $\log \sigma$ (LogSigmoid) in the following form:

$$
\mathcal{V}_{\text{LS}} = \mathbb{E}_{p_{\text{data}}(y)p(x|y)q(x'|y)}[\log \sigma(\tilde{f}_1(x) - \tilde{f}_2(x'))] \tag{41}
$$

Specifically, $\mathcal{V}_{\text{BT-C}}$ is recovered by setting $\tilde{f}_1 = \tilde{f}_2 = f$, $p = p_{\text{data}}(\cdot|y)$, and $q = p_g(\cdot|y)$ in Equation (41), while $\mathcal{V}_{\text{BT-M}}$ is obtained by setting $\tilde{f}_1 = \tilde{f}_2 = f$, $p = p_{\text{data}}(\cdot|y)$, and $q = p_{\text{data}}$.

Recall also that we adopt $\mathcal{V}_{\text{ORIG-GAN}}$ proposed in Goodfellow et al. (2014) (see Equation (18) in Appendix B.1) to define $\mathcal{V}_{\text{SAN}}$, which is specifically formulated as:

$$
\mathcal{V}_{\text{SAN}} = \mathbb{E}_{p_{\text{data}}(x)}[\log \sigma(f_{\Phi_N}^{\text{N}}(x))] + \mathbb{E}_{p_g(x)}[\log \sigma(-f_{\Phi_N}^{\text{N}}(x))] \tag{42}
$$

$$
= \mathbb{E}_{p_{\text{data}}(x)}[\log \sigma(\langle \omega, h(x) \rangle - (-b))] + \mathbb{E}_{p_g(x)}[\log \sigma(-b - \langle \omega, h(x) \rangle)], \tag{43}
$$

where $\mathcal{V}_{\text{SAN}}$ includes two terms involving $\log \sigma(\cdot)$, each recovered by setting $(p, q, \tilde{f}_1, \tilde{f}_2) = (p_{\text{data}}, p_{\text{data}}, \langle \omega, h \rangle, -b)$ and $(p, q, \tilde{f}_1, \tilde{f}_2) = (p_g, p_g, -b, \langle \omega, h \rangle)$ in Equation (41), respectively.

As proposed in Section 4.4, we replace $\log \sigma(t)$ with $\log \sigma(s \cdot t)/s$ in Equation (41), yielding:

$$
\mathcal{V}_{\text{LS},s} = \mathbb{E}_{p_{\text{data}}(y)p(x|y)q(x'|y)}\left[\frac{1}{s}\log \sigma\left(s(\tilde{f}_1(x) - \tilde{f}_2(x'))\right)\right] \tag{44}
$$

For simplicity, we consider a single update step for $s$ and a one-sample approximation of the expectation (for $\mathcal{V}_{\text{ORIG-GAN}}$, we stochastically compute either the first or second term in Equation (43) per iteration). This leads to:

$$
\tilde{\mathcal{V}}_{\text{LS},s} = \frac{1}{s}\log \sigma(s(\underbrace{\tilde{f}_1(x) - \tilde{f}_2(x')}_{\Delta \tilde{f}})), \tag{45}
$$

where $x \sim p$ and $x' \sim q$. The derivative of $\tilde{\mathcal{V}}_{\text{LS},s}$ with respect to $s$ is:

$$\frac{\partial \tilde{\mathcal{V}}_{\text{LS},s}}{\partial s} = \frac{\partial}{\partial s} \left[ \frac{1}{s} \log \sigma(s\Delta\tilde{f}) \right] \tag{46}$$

$$= \frac{1}{s^2} \left( \frac{s\Delta\tilde{f}}{\exp(s\Delta\tilde{f}) + 1} - \log\sigma(s\Delta\tilde{f}) \right) \tag{47}$$

For $0 < s < 1$ (from the constraint on $s$), this derivative $\partial\tilde{\mathcal{V}}_{\text{LS},s}/\partial s$ has the following properties:

(P1) For fixed $0 < s < 1$ and any $\Delta\tilde{f}$, it is monotonically increasing with respect to $\Delta\tilde{f}$.

(P2) For fixed $\Delta\tilde{f} \geq 0$, it is monotonically decreasing with respect to $s$.

To illustrate these properties, consider the two-term case:

$$\frac{1}{s_1} \log\sigma(s_1\Delta\tilde{f}_1) + \frac{1}{s_2} \log\sigma(s_2\Delta\tilde{f}_2) \tag{48}$$

In this setup, (P1) implies that when $0 < s_1 = s_2 < 1$, the coefficient corresponding to the larger error between $\Delta\tilde{f}_1$ and $\Delta\tilde{f}_2$ yields a larger gradient, meaning the larger error is prioritized by increasing its coefficient. (P2) implies that when $\Delta\tilde{f}_1 = \Delta\tilde{f}_2 \geq 0$, the smaller of $s_1$ and $s_2$ has a larger gradient, leading to $s_1 = s_2$ if this equality persists during optimization.

Kendall et al. (2018) proposed an adaptive weighting scheme that introduces a scalar parameter to represent uncertainty. Although this method shares some similarities with ours, their method balances multiple terms using (learnable) unbounded coefficients, which can diverge as training progresses. This unbounded growth is undesirable in our case, as GAN training stability is generally sensitive to the learning rate.

# F  EXPERIMENTAL DETAILS

## F.1  MoG EXPERIMENTS IN SECTION 5.2

We empirically evaluate our proposed method in a two-dimensional space $X = \mathbb{R}^2$. The target mixture of Gaussians (MoG) in $X$ consists of $N$ isotropic Gaussian components, each with variance $0.03^2$ and means evenly distributed on a circle of radius 0.75. The generator is modeled with a 10-dimensional latent space $Z$, where the base distribution $p(z)$ is standard normal.

Both the generator and discriminator use simple fully connected (FC) architectures, following previous work (Mescheder et al., 2017; Nagarajan & Kolter, 2017; Sinha et al., 2020; Takida et al., 2024). Specifically, each network consists of three hidden FC layers with 50 units per layer. The generator uses ReLU activations, while the discriminator uses Leaky ReLU, which facilitates the discriminator's injectivity (Takida et al., 2024). The last linear layer in the discriminator corresponds to $\omega$ and $w$ for SONA and PD-GAN, respectively. For class conditioning in the generator, we use four-dimensional learnable class embeddings concatenated with the input noise $z$. For the discriminator, we use additional class-dependent embeddings: $w_y$ for PD-GAN and $\omega_y$ for SONA. In SONA, both the linear projection and the embeddings in the discriminator are normalized to ensure $\omega, \omega_y \in \mathbb{S}^{50-1}$.

For training, we use a batch size of 256 and the Adam optimizer (Kingma & Ba, 2015) with $(\beta_1, \beta_2) = (0.0, 0.9)$ and learning rates of 0.0001 for both the generator and discriminator. The update ratio is set to 1, meaning the discriminator is updated once per iteration. Models are trained for 15,000 iterations, and the checkpoint with the lowest **W2** value is selected as the best model.

Wasserstein-2 distances for **W2**, **cW2**, and **NF** are computed using the POT toolbox[5] (Flamary et al., 2021; 2024) with 10,000 samples per distribution.

---

[5] https://github.com/PythonOT/POT

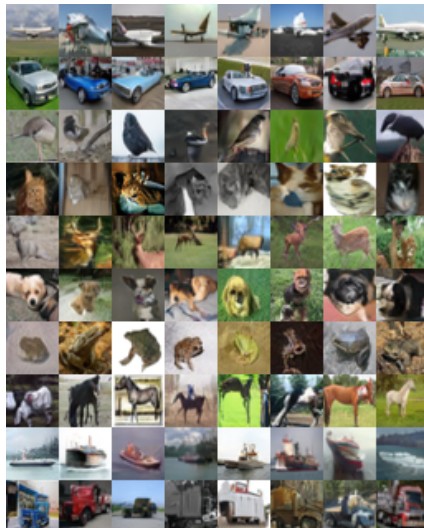 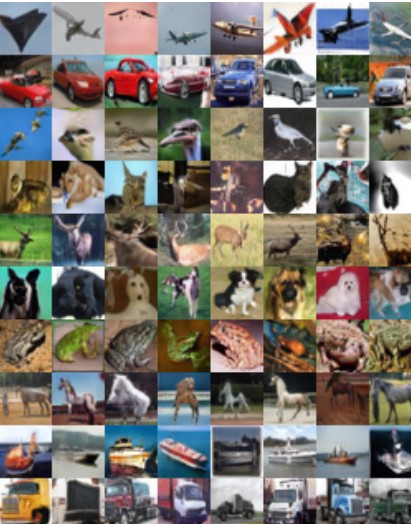

Figure 5: CIFAR10: (Left) Generated samples by SONA with BigGAN. (Right) Generated samples by SONA with StyleGAN-2.

## F.2 Class Conditional Generation Tasks in Section 6.1

### F.2.1 Experimental Setup

We base our experiments on the benchmarking repository provided by PyTorch-StudioGAN (Kang et al., 2023a). For hyperparameters such as learning rate and batch size, we strictly follow the default configuration provided for PD-GAN.

To ensure fair comparisons, we conduct all experiments ourselves and report the resulting scores in the tables. All models are trained on CIFAR10 and TinyImageNet three times with different random seeds; we report the mean and standard deviation of the scores in Tables 2 and 3. For ImageNet, due to the high computational cost (each training run requires 8 and 40 H100-days for batch sizes of 256 and 2048, respectively), we report results from a single run.

For baselines, we select two representative classifier-based methods, ContraGAN and ACGAN, and one projection-based method, PD-GAN. Since our primary objective is to compare SONA with other state-of-the-art discriminator conditioning methods, we do not include additional data augmentation (Karras et al., 2020a; Zhao et al., 2020) or discriminator regularization techniques (Zhang et al., 2020; Zhao et al., 2021; Tseng et al., 2021), as these are orthogonal to our approach. To demonstrate that our method can be combined with such techniques, we also compare SONA and the baselines using the DiffAug data augmentation method (Zhao et al., 2020), and confirm that the performance of SONA can be further improved, as shown in Table 3.

## F.3 Text-to-Image Generation Tasks in Section 6.2

### F.3.1 Experimental setup

We base our experiments on the benchmarking repository provided by Kang et al. (2023a). For hyperparameters such as learning rate and batch size, we strictly follow the default configuration.

### F.3.2 Converting GALIP discriminator with SONA

**Discriminator architecture.** We briefly review the discriminator architecture proposed by Tao et al. (2023), which serves as our base architecture. The GALIP discriminator consists of a frozen CLIP-ViT and a learnable module called Mate-D. Mate-D is designed to effectively utilize deep features extracted from both images and text using CLIP. Specifically, Mate-D comprises a CLIP Feature Extractor (CLIP-FE) and a quality assessor (QA). The CLIP-FE aggregates multi-layer features from CLIP-ViT using a sequence of extraction blocks, each containing convolutional and ReLU

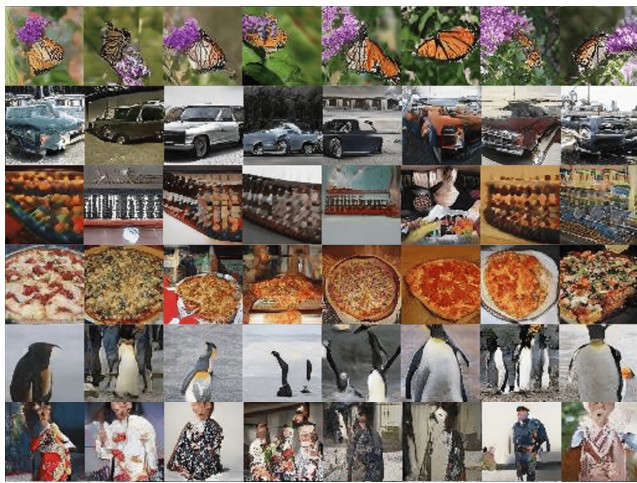

Figure 6: TinyImageNet: Generated samples by SONA applied with DiffAug.

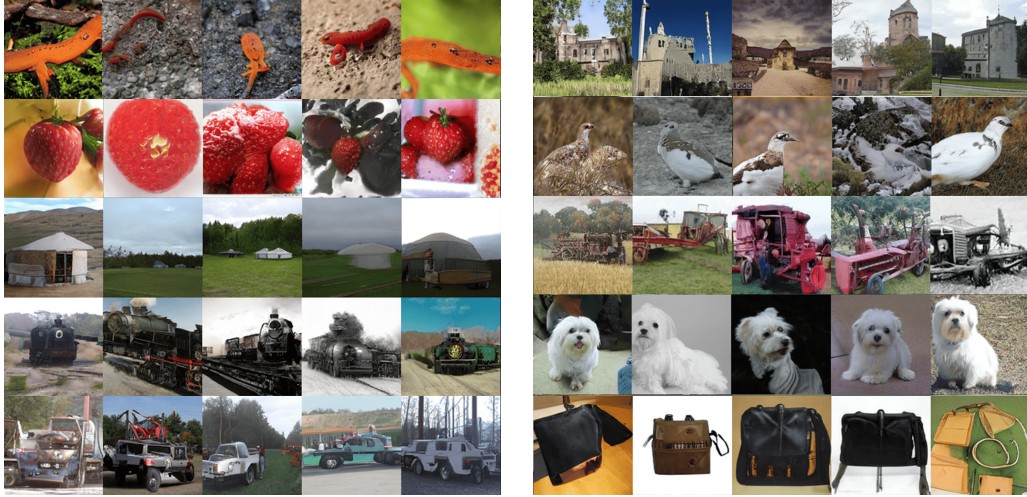

Figure 7: ImageNet: Generated samples by SONA trained with batch size of 2048.

layers, to progressively refine visual representations. The final extracted features are concatenated with replicated sentence vectors obtained by feeding text prompts ($y$) into the CLIP text encoder. These concatenated features are then evaluated by the QA, which predicts conditional likelihood using additional shallow convolutional layers to assess image quality. The dimensionality of the final extracted features, corresponding to $h(x)$ in our formulation, and the sentence vector, denoted as $e(y)$, are 512 and $512 \times 7 \times 7$, respectively. The QA converts $e(y) \in \mathbb{R}^{512}$ to $E(y) \in \mathbb{R}^{512 \times 7 \times 7}$ by replicating the 512-dimensional vector 49 times to enable concatenation.

Applying SONA to this discriminator requires only modifications to the QA. We add a single trainable $512 \times 7 \times 7$-dimensional parameter to model $\omega$. For $\omega_y$, we directly use the extended text embeddings $E(y)$. Notably, even though $\omega_y$ is frozen in this setup, SONA achieves improved generation performance and comparable text alignment. We also experimented with modeling $\omega_y$ using learnable modules, such as a learnable affine layer or a shallow FC network applied to $e(y)$ to produce a $512 \times 7 \times 7$-dimensional feature. However, these approaches degraded performance, particularly in text alignment. We suspect that applying such learnable operators to CLIP features may cause information loss and prevent full utilization of the pre-trained representations without careful design. Designing suitable modules for $\omega_y$ based on CLIP features remains an open direction for future work.

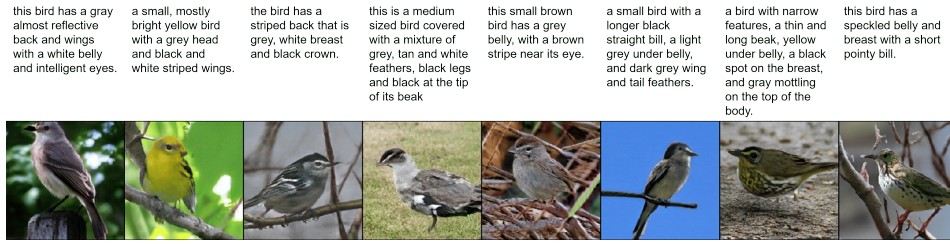

Figure 8: CUB: Generated samples by SONA.

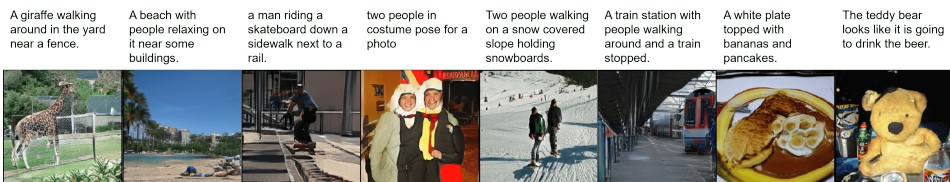

Figure 9: COCO: Generated samples by SONA.

**Training objective.** GALIP incorporates additional objective terms and techniques into both the discriminator and generator losses to enhance text alignment.

For the discriminator, the fake distribution (i.e., the generator distribution in standard GANs) is augmented with a mixture distribution that combines the generator distribution and a mismatched data distribution, formed by incorrect image-text pairs in equal proportion. To further stabilize adversarial training, a matching-aware gradient penalty (MAGP) is applied to both the extracted CLIP features and their corresponding text features. For the generator, a CLIP-based cosine similarity loss is added to encourage both image quality and text alignment. The overall objective functions are given by

$$\mathcal{V}_{\text{GALIP}}(f) = \mathbb{E}_{p_{\text{data}}(x_{\text{d}},y)}[\min(0, -1 + f(x_{\text{d}}, y))] + \frac{1}{2}\mathbb{E}_{p_g(x_g)}[\min(0, -1 - f(x_g, y))] \tag{49}$$

$$+ \frac{1}{2}\mathbb{E}_{p_{\text{data}}(x_{\text{d}})p_{\text{data}}(y)}[\min(0, -1 - f(x_{\text{d}}, y))] + \lambda_1 \text{MAGP} \tag{50}$$

$$\mathcal{J}_{\text{GALIP}}(g) = -\mathbb{E}_{p_g(x_g|y)p_{\text{data}}(y)}[f(x_g, y)] - \lambda_2 \mathbb{E}_{p_g(x_g|y)p_{\text{data}}(y)}[S_{\text{CLIP}}(x_g, y)], \tag{51}$$

where $S_{\text{CLIP}}$ denotes the CLIP-based cosine similarity.

For a fair comparison, we partially follow the original loss by adding MAGP to the discriminator loss and including the same CLIP-based similarity loss in the generator objective. For the remaining loss terms, SONA provides direct counterparts, which replace the original objectives. We use exactly the same values of $\lambda_1$ and $\lambda_2$.

## G GENERATED SAMPLES

Generated samples by SONA trained in Section 6 can be found in Figures 5 to 10.

## H LIMITATIONS AND FUTURE WORKS

**Method.** Our method is, in principle, applicable to a wide range of conditional generation tasks. However, efficiently modeling conditional projections $\omega_y$ is still challenging when $Y$ is not a finite discrete set (e.g., when $Y$ consists of plausible text captions or is continuous). In our text-to-image experiments (Section 6.2), we use frozen embeddings from a pre-trained CLIP encoder, which may limit the discriminator's representational power for conditional alignment. Developing effective approaches for modeling conditional projections that generalize to arbitrary types of $Y$ remains an open problem.

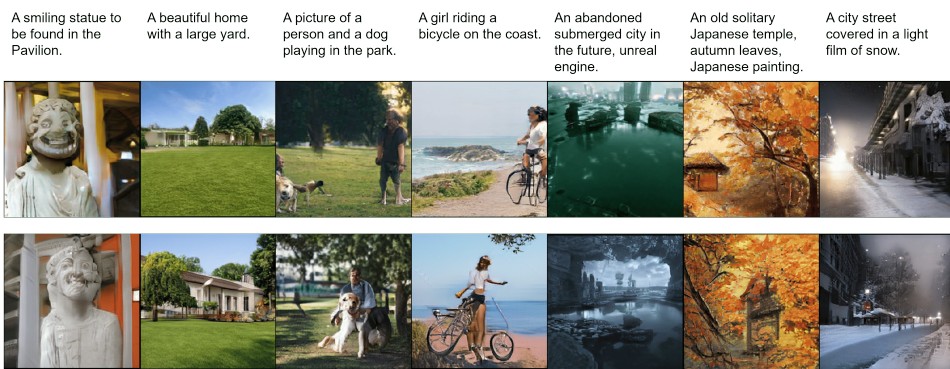

Figure 10: GALIP models trained on CC12M: (Top) Generated samples by GALIP (concat). (Bottom) Generated samples by GALIP with SONA. Text prompts are from the COCO dataset.

**Theoretical Analysis.** In Proposition 3, we assume the discriminator is globally optimal. While this assumption is common in the literature (Goodfellow et al., 2014; Johnson & Zhang, 2019; Gao et al., 2019; Fan et al., 2022; Li et al., 2018; Chu et al., 2020), it rarely holds in practical GAN optimization. Extending the theoretical analysis to more relaxed and realistic conditions on the discriminator is an important direction for future work.

**Experiments.** We evaluated our approach on standard benchmarks with images up to $256 \times 256$ resolution, addressing both class- and text-conditional generation tasks. Expanding to a wider range of conditioning modalities (e.g., segmentation maps, image style) and larger-scale settings (e.g., $512 \times 512$ or higher, progressive learning setups (Sauer et al., 2022)), as well as extending beyond image generation to domains such as video and audio generation, are important directions for future research.

**Future Works beyond GANs.** Our scope is discriminators, which are beneficial general generative models beyond GANs, including diffusion models. One of the most active areas in this line is diffusion distillation into one-step or few-step generative models. Adversarial Diffusion Distillation (Sauer et al., 2024, ADD) is a pioneering paper of this direction, in which adversarial loss based on a discriminator is used altogether with the usual distillation loss, enhancing the distillation performance significantly. ADD is a backbone framework used for training SDXL Turbo, a well-known high-quality text-to-image model. Besides ADD, some work also employs adversarial training to enable the generation of high-quality samples with one step (Kang et al., 2024; Lin et al., 2025; Chen et al., 2025a). According to this literature, improving discriminators can potentially lead to improved diffusion-based generative models. While it is an interesting trial to apply SONA to such distillation methods, it represents a substantial topic for future investigation.

