# OpenReview forum: "SONA: Learning Conditional, Unconditional, and Matching-Aware Discriminator"
_ICLR.cc/2026/Conference — ICLR 2026 Poster_

### Official Review · Reviewer_nbxK · 2025-10-31

**Soundness:** 4
**Presentation:** 3
**Contribution:** 3
**Rating:** 6
**Confidence:** 4

**Summary:**

This paper introduces SONA (Sum of Naturalness and Alignment), a new discriminator framework for conditional generative adversarial networks (GANs). The motivation stems from a persistent issue in conditional generation: balancing sample authenticity (unconditional discrimination) with conditional alignment (matching the conditioning signal such as class labels or text prompts). Traditional discriminators, such as those in AC-GAN or projection-based GANs (PD-GAN), handle these two aspects in an intertwined manner, often requiring careful weighting or suffering from limited sensitivity to mismatched conditions.

SONA explicitly decomposes the discriminator’s learning objective into three coordinated components:

1. Unconditional discrimination, focusing on the authenticity of generated samples via the Sliced Wasserstein perspective;

2. Matching-aware discrimination, improving alignment sensitivity by introducing mismatched (negative) samples through a Bradley–Terry pairwise comparison formulation;

3. Adaptive weighting, dynamically balancing the above objectives through learnable coefficients constrained by normalization.

The proposed design uses orthogonal projections of discriminator features to separately assess naturalness and alignment, encoding an inductive bias that disentangles these two factors. The theoretical analysis establishes that each component corresponds to optimizing distinct divergences (e.g., log-probability gaps between conditional and unconditional distributions).

Empirically, SONA achieves consistent gains over state-of-the-art baselines (ReACGAN, ContraGAN, PD-GAN) on CIFAR10, TinyImageNet, and ImageNet, improving both FID and Inception Score across multiple backbones (BigGAN, StyleGAN2). The method further extends effectively to text-to-image generation tasks (CUB, COCO, CC12M) when integrated with GALIP, achieving better FID and comparable CLIP alignment. Ablation studies confirm that orthogonal modeling and mismatching-aware loss significantly contribute to performance improvements.

Overall, SONA presents a unified, theoretically sound, and empirically validated framework for improving conditional discriminators, offering both conceptual clarity and practical robustness in modern GAN training.

**Strengths:**

1. Novel conceptual integration: The paper elegantly unifies multiple important discriminator desiderata—unconditional, conditional, and mismatching-aware discrimination—into one coherent framework. The separation of naturalness and alignment projections via orthogonality is an intuitively appealing and theoretically well-justified idea.

2. Sound theoretical grounding: Each component of SONA is supported by propositions that connect its loss functions to meaningful probabilistic interpretations (e.g., conditional divergence minimization and log probability gap maximization). The use of the Bradley–Terry model for conditional discrimination is a clever adaptation rarely seen in GAN literature.

3. Strong empirical performance: The method demonstrates clear and consistent improvements across diverse datasets and architectures. Especially on ImageNet (FID=6.14, IS=140.14), SONA substantially outperforms prior methods, showing robustness at scale. The inclusion of text-to-image benchmarks further validates its generality.

4. Comprehensive experimentation: The authors conduct detailed comparisons with strong baselines (PD-GAN, ReACGAN, ContraGAN), extensive ablations (adaptive weighting, orthogonal projection, mismatch loss), and visualization using toy datasets (MoG). This provides convincing evidence of both quantitative and qualitative improvements.

5. Reproducibility and transparency: The paper references open-source implementations (StudioGAN, BigGAN-PyTorch, GALIP) and commits to code release. Appendices include detailed algorithmic descriptions and theoretical proofs.

6. Practical relevance: Since projection-based discriminators remain a backbone of modern GANs, SONA’s design can be easily adopted without significant architectural overhead, making it both theoretically elegant and engineering-friendly.

Overall, the paper succeeds in addressing a long-standing problem (balancing authenticity and alignment) in a way that is theoretically principled, empirically validated, and broadly applicable across generative modalities.

**Weaknesses:**

1. Complexity of formulation: While conceptually clean, the proposed method involves multiple interrelated objectives (SAN, BT-COND, BT-MM, adaptive weighting), which may make training more cumbersome in practice. Although the authors report stability, more discussion on convergence behavior and computational overhead (especially in large-scale settings) would strengthen the paper.

2. Limited comparison with diffusion-based models: The field has largely shifted toward diffusion and rectified flow architectures for conditional generation (e.g., SDXL, RF Transformers). While SONA’s GAN focus is clear, the paper could better contextualize its relevance in the broader landscape of generative modeling in 2025–2026, potentially by discussing hybrid or complementary uses.

3. Ablation depth: While ablation studies isolate major components, finer-grained analysis (e.g., sensitivity of adaptive weighting parameters or orthogonality constraint strength) is missing. It would also be useful to examine cases where SONA may underperform or produce mode collapse, to better understand its limitations.

4. Clarity and accessibility: The theoretical derivations, especially those involving the Sliced Wasserstein and Bradley-Terry frameworks, are mathematically dense and may be challenging for readers not familiar with those formulations. A more intuitive explanation or schematic of the optimization flow could enhance accessibility.

**Questions:**

1. Computational overhead: How much additional computation or memory does SONA introduce compared to standard PD-GANs, particularly due to pairwise BT loss sampling and adaptive weighting? Is the training speed comparable?

2. Stability and convergence: Theoretical results show desirable properties at equilibrium, but how stable is the actual training dynamic? Did the authors observe any oscillations or mode collapse before convergence?

3. Generalizability to diffusion models: Since diffusion-based models dominate text-to-image generation, can SONA’s discriminator concept be repurposed as a “critic” for conditional diffusion or consistency models?

4. Scaling to complex conditions: How does SONA perform when the conditional space is continuous or high-dimensional (e.g., embeddings or attributes rather than discrete labels)?

---

> ### Author Response · Authors · 2025-11-21
> **Rebuttal Reply to Reviewer nbxK (1/n)**
>
> Thank you for your review. Please kindly find our reply below.
>
> ---
>
> ### [n-1] SONA is simpler than classifier-based approaches in practice
>
> > Complexity of formulation: While conceptually clean, the proposed method involves multiple interrelated objectives (SAN, BT-COND, BT-MM, adaptive weighting), which may make training more cumbersome in practice.
>
> Thank you for your valuable comment, which allows us to highlight the advantages of our adaptive weighting method and to further clarify why our approach is computationally efficient compared with classifier-based methods, as empirically demonstrated in Table 7. Please find our detailed response below. We have **updated Appendix A** to discuss this perspective accordingly.
>
> We would like to emphasize that our proposed method is more user-friendly for practitioners than classifier-based methods, for the following two reasons. First, our method does not require additional hyperparameters that need to be tuned in advance beyond those used in unconditional setups, which is a benefit shared only with projection-based methods. Second, our method is computationally efficient compared with classifier-based methods. We elaborate on each point below.
>
> **(1) No need for hyperparameter tuning:** Although our proposed discriminator loss consists of multiple objective terms, the learnable coefficients described in Section 4.4 automatically balance these objectives, removing the necessity for manual hyperparameter tuning. Classifier-based methods also employ discriminator losses with multiple objectives, but these are balanced by manually tuned coefficients. In this respect, our proposed loss is more user-friendly for practitioners than conventional classifier-based approaches.
>
> **(2) Computational efficiency compared with classifier-based methods:** From an architectural perspective, SONA requires additional linear projections $\omega_y$ to unconditional discriminators, which is also necessary for general conditional discriminators, including both projection- and classifier-based approaches. Regarding the objective function, the first term in Equation (12), $\mathcal{V}\_{\text{SAN}}$, is nearly identical in complexity to the loss used in projection-based methods and the unconditional term in classifier-based approaches. The second and third terms in Equation (12), $\mathcal{V}\_{\text{BT-C}}$ and $\mathcal{V}\_{\text{BT-M}}$, require a losing sample per training sample, which entails additional computation of discriminator outputs for these losing samples. However, this computation is less expensive compared to the classification loss in classifier-based methods. This difference can be explained by considering the InfoNCE-loss perspective, which is explained around Proposition 4 in Section 5. The objective terms for SONA can be interpreted as InfoNCE losses using only a single negative sample $x’$ drawn from $p_{\text{d}}(x’)$ per positive (training) sample $x$ drawn from $p_{\text{d}}(x|y)$, whereas classifier-based methods use an InfoNCE loss that considers all plausible negative labels $y’\sim p_{\text{d}}(y’)$ for each positive (training) sample $y\sim p_{\text{d}}(y|x)$. This characteristic of classifier-based approaches also makes them less applicable to text-to-image scenarios, where the set of plausible labels is not finite, as there are infinitely many possible text prompts.
>
> In summary, SONA is more computationally efficient than classifier-based methods, as it introduces contrastive objective terms with a minimal number of negative samples. This fact is consistent with the results in Table 7, which reports the computational complexity of training for each method.

---

> > ### Author Response · Authors · 2025-11-21
> > **Rebuttal Reply to Reviewer nbxK (2/n)**
> >
> > ### [n-2] Our new plots show the convergence behavior of SONA and the benefit of SONA compared to baselines in computational–performance trade-off
> >
> > > Although the authors report stability, more discussion on convergence behavior and computational overhead (especially in large-scale settings) would strengthen the paper.
> >
> > > Computational overhead: How much additional computation or memory does SONA introduce compared to standard PD-GANs, particularly due to pairwise BT loss sampling and adaptive weighting? Is the training speed comparable?
> >
> > Thank you for your constructive suggestion. In response, we have **added a new subsection (Section 6.4)** dedicated to discussing SONA from these perspectives. In particular, we have **included a new figure** to further compare our method with PD-GAN in terms of the computational–performance trade-off in the revised manuscript (see **Figure A**). Please find our detailed response below.
> >
> > The initial manuscript reports that SONA is slower than PD-GAN in training time per iteration, while it is faster than classifier-based methods such as ContraGAN and ReACGAN, as shown in Table 7. To further discuss convergence behavior and justify the computational overhead of SONA, we have included a new plot illustrating the relationship between **training time** and FID/IS based on experiments on ImageNet with a batch size of 2048, which is the largest class-conditional setting (Figure A). This plot enables comparison of SONA with the baselines from the perspective of computational–performance trade-off. The results indicate that SONA achieves FID scores comparable to those of PD-GAN during approximately the first three days of training. After this period, SONA demonstrates a clear improvement, ultimately achieving lower FID scores. Notably, SONA attains a better FID score than the best result achieved by PD-GAN, and does so with a shorter training duration. Therefore, this new plot highlights the advantages of SONA over PD-GAN in terms of training efficiency.
> >
> > > Stability and convergence: Theoretical results show desirable properties at equilibrium, but how stable is the actual training dynamic? Did the authors observe any oscillations or mode collapse before convergence?
> >
> > We did not observe instability issues in our experiments. Additionally, as shown in Figure A, the scores for SONA improve almost monotonically and smoothly during training, which supports the stable behavior of SONA training.

---

> > > ### Author Response · Authors · 2025-11-21
> > > **Rebuttal Reply to Reviewer nbxK (3/n)**
> > >
> > > ### [n-3] We have added a description on the benefit of our general idea for generative models beyond GANs
> > >
> > >
> > > > Limited comparison with diffusion-based models: The field has largely shifted toward diffusion and rectified flow architectures for conditional generation (e.g., SDXL, RF Transformers). While SONA’s GAN focus is clear, the paper could better contextualize its relevance in the broader landscape of generative modeling in 2025–2026, potentially by discussing hybrid or complementary uses.
> > >
> > > > Generalizability to diffusion models: Since diffusion-based models dominate text-to-image generation, can SONA’s discriminator concept be repurposed as a “critic” for conditional diffusion or consistency models?
> > >
> > > Thank you for your constructive suggestions, which are very helpful for better positioning our method in relation to modern generative models. As you mentioned, our work could be potentially beneficial for modern generative models beyond GANs, as detailed below. We have **added the following discussion to Appendix H** in the revision.
> > >
> > > Because our scope is discriminators for general adversarial generative modeling setups, we believe that it is beneficial for a broader class of generative models, including diffusion models. One of the most active areas in this line is diffusion distillation into one-step or few-step generative models. Adversarial Diffusion Distillation (AAD; Sauer+2023) is a pioneering paper of this direction, in which adversarial loss based on a discriminator is used altogether with the usual distillation loss, enhancing the distillation performance significantly. AAD is a backbone framework used for training SDXL Turbo, a well-known high-quality text-to-image model. Besides AAD, some work also employs adversarial training to enable the generation of high-quality samples with one step (e.g., [Kang+2024] and [Lin+2025]). According to this literature, improving discriminators can potentially lead to improved diffusion-based generative models. While it is an interesting trial to apply SONA to such distillation methods, there is no public code for such works to the best of our knowledge. Instead, this topic would be left as future work at this moment.
> > >
> > > Additionally, to contextualize SONA in current generative models in terms of performance, we have updated the table reporting the results of text-to-image generation tasks under the zero-shot setting on COCO (Table 5-B in the revised manuscript).
> > >
> > > [Sauer+2023] “Adversarial Diffusion Distillation,” ECCV2024
> > >
> > > [Kang+2024] “Distilling Diffusion Models into Conditional GANs,” ECCV2024
> > >
> > > [Lin+2025] “Diffusion Adversarial Post-Training for One-Step Video Generation,” ICML2025
> > >
> > > [Xu+2025] “VideoGigaGAN: Towards Detail-rich Video Super-Resolution,” CVPR2025
> > >
> > > ---
> > >
> > > ### [n-4] SONA does not include any hyperparameters that require tuning in advance beyond those used in unconditional setups
> > >
> > > > Ablation depth: While ablation studies isolate major components, finer-grained analysis (e.g., sensitivity of adaptive weighting parameters or orthogonality constraint strength) is missing.
> > >
> > > SONA does not include any hyperparameters for the adaptive weighting and orthogonal constraint. Therefore, our ablation study is limited to isolating each component, without manual parameter sweeping. Specifically, the coefficients for balancing multiple discriminator objectives are fully learnable, and the inductive bias for direction orthogonality is implemented using a simple pre-defined orthogonal projection.
> > >
> > > ---
> > >
> > > ### [n-5] We have Appendix H, a dedicated section for discussing limitations
> > >
> > > > It would also be useful to examine cases where SONA may underperform or produce mode collapse, to better understand its limitations.
> > >
> > > We acknowledge the importance of discussing potential limitations. Our initial manuscript includes a section dedicated to the limitations of this work (see Appendix H). However, as we did not observe any cases where SONA resulted in significant underperformance or mode collapse, we have focused our discussion of limitations on methodological, theoretical, and experimental perspectives.

---

> > > > ### Author Response · Authors · 2025-11-21
> > > > **Rebuttal Reply to Reviewer nbxK (4/4)**
> > > >
> > > > ### [n-6] We have added two subsections to Appendix B, which review explanation of the Sliced Wasserstein concept and the Bradley-Terry framework, respectively
> > > >
> > > > > Clarity and accessibility: The theoretical derivations, especially those involving the Sliced Wasserstein and Bradley-Terry frameworks, are mathematically dense and may be challenging for readers not familiar with those formulations. A more intuitive explanation or schematic of the optimization flow could enhance accessibility.
> > > >
> > > > Thank you for your helpful suggestion. We agree that a clearer introduction to the preliminary concepts will benefit a broader range of readers. In response, we have **expanded Appendix B** by including a more detailed explanation of SAN in a new subsection and adding a dedicated subsection on the Bradley-Terry framework. We believe these additions in Appendix B will help readers who are unfamiliar with these concepts.
> > > >
> > > > ---
> > > >
> > > > ### [n-7] CLIP embedding, which is used in GALIP for text-to-image generation, is continuous and high-dimensional
> > > >
> > > > > Scaling to complex conditions: How does SONA perform when the conditional space is continuous or high-dimensional (e.g., embeddings or attributes rather than discrete labels)?
> > > >
> > > > In Section 6.2, we applied our method to text-to-image generation based on GALIP to demonstrate its applicability beyond class-conditional setups. We believe that the experiments involving text-to-image generation address this comment, since the CLIP embeddings used to represent text conditions are continuous and high-dimensional (512-dimensional). Although applying our method to cases with even more complex conditioning, such as higher-dimensional conditional spaces, would be an interesting direction for future research, we consider that the current experiments provide valuable insights for such future applications.

---

### Official Review · Reviewer_xeHQ · 2025-11-03

**Soundness:** 3
**Presentation:** 3
**Contribution:** 2
**Rating:** 4
**Confidence:** 3

**Summary:**

The paper addresses a key limitation in conditional GANs, which often struggle to jointly ensure image authenticity and alignment with conditioning inputs. To mitigate this, they propose SONA, a novel discriminator architecture that integrates unconditional discrimination, matching-aware supervision, and adaptive weighting. Experiments on class-conditional and text-to-image generation tasks demonstrate that SONA achieves superior sample quality and conditional alignment compared to existing methods.

**Strengths:**

* The paper includes comprehensive experiments conducted across diverse datasets and scales, supporting the generality of the proposed method.
* The paper includes an ablation study that justifies the contribution of each model component. It would be helpful to include visualizations to show how each component affects output quality.
* The writing is clear and well-organized.

**Weaknesses:**

* The relevance of the addressed problem is uncertain in the current context, where most major advances in generative modeling stem from diffusion models rather than GANs. Moreover, the compared baselines are relatively outdated GAN architectures, making the reported improvements less meaningful. It remains unclear whether the proposed approach would offer advantages over using state-of-the-art diffusion-based methods, which limits the overall impact of the work.
* The paper lacks baseline comparisons with modern diffusion models, which currently outperform GANs on most conditional generation benchmarks.
* Missing qualitative results. Given that the task involves image generation, it should include visual comparisons illustrating how the proposed method improves sample quality and conditional alignment relative to existing approaches.

**Questions:**

Mentioned in the weakness section.

---

> ### Author Response · Authors · 2025-11-21
> **Rebuttal Reply to Reviewer xeHQ (1/n)**
>
> Thank you for your valuable comments. Please kindly find our reply below.
>
> ---
>
> ## General comment regarding our contributions
>
> Before addressing each specific review comment, we would like to highlight our technical contributions. We have developed a framework for an effective conditional discriminator that incorporates unconditional, conditional, and matching-aware discrimination, with adaptive weighting for each. Our technical contribution has been recognized by Reviewers ip5t, cYPq, and nbxK. Additionally, our extensive experiments serve as valid demonstrations that our method is broadly applicable for improving conditional discriminators, as noted by Reviewers ip5t, cYPq, and nbxK. We believe that our work makes significant contributions to the advancement of general conditional discriminators. Furthermore, it has the potential to benefit future research in generative modeling, including not only the enhancement of GANs but also the improvement of diffusion distillation (please refer to our reply to [x-1]). We will elaborate further by addressing each comment below. We would appreciate it if you could reconsider the position and value of our paper.

---

> > ### Author Response · Authors · 2025-11-21
> > **Rebuttal Reply to Reviewer xeHQ (2/n)**
> >
> > ### [x-1] Our general idea could be beneficial for general generative models beyond GAN
> >
> > > The relevance of the addressed problem is uncertain in the current context, where most major advances in generative modeling stem from diffusion models rather than GANs….. It remains unclear whether the proposed approach would offer advantages over using state-of-the-art diffusion-based methods, which limits the overall impact of the work.
> >
> > Thank you for your comment, which is helpful for better positioning our method in relation to modern generative models. We would like to address this comment in two fold below. In short, we believe our work can potentially be beneficial for (1) enhancement of diffusion-based models, and (2) development of greater GANs, in the future. We have **added the first point to Appendix H** in the revised manuscript as a direction for future work.
> >
> > First, our scope is discriminators, which are beneficial general generative models beyond GANs, including diffusion models. One of the most active areas in this line is diffusion distillation into one-step or few-step generative models. Adversarial Diffusion Distillation (ADD; Sauer+2023) is a pioneering paper of this direction, in which adversarial loss based on a discriminator is used altogether with the usual distillation loss, enhancing the distillation performance significantly. ADD is a backbone framework used for training SDXL Turbo, a well-known high-quality text-to-image model. Besides ADD, some work also employs adversarial training to enable the generation of high-quality samples with one step (e.g., [Kang+2024], [Lin+2025] and [Chen+2025]). According to this literature, improving discriminators can potentially lead to improved diffusion-based generative models.
> >
> > Although adversarial approaches for diffusion distillation have demonstrated strong performance and SONA could potentially offer further improvements, several open research challenges remain. Most existing methods combine adversarial and regression losses, which requires careful balancing of these objectives. Our adaptive weighting for SONA removes the need for hyperparameter tuning when balancing multiple objectives in conditional GANs, but further development is needed to establish principled methods for adaptively balancing all terms, including the regression loss, to fully leverage the benefits of our approach in this context. There is also ongoing discussion regarding the necessity of the regression loss itself; for example, [Lin+2025] omit this term. Thus, while applying SONA to diffusion distillation is a promising direction, it represents a substantial topic for future investigation. Furthermore, a straightforward application of SONA is not feasible in this short discussion period, as there is no public code for these works to the best of our knowledge.
> >
> > Second, GAN is still a useful family of deep generative models even as standalone models. One of the biggest benefits of GANs is the speed of sampling. Generally, generators trained by the GAN framework are much faster than diffusion models, which typically require hundreds to thousands of inference steps in generation. For example, GANs have demonstrated impressive performance in text-to-image generation. GigaGAN, for instance, achieves results that are competitive with or superior to those of Stable Diffusion v1.5 (with a similar parameter size), while requiring less than one-twentieth of the inference time (Kang+ 2023). Furthermore, GANs exhibit great performance in specific generative tasks such as super-resolution (Xu+2025) in vision and vocoding (Lee+2023) in audio.
> >
> > [Lee+2023] “BigVGAN: A Universal Neural Vocoder with Large-Scale Training,” ICLR2023
> >
> > [Sauer+2023] “Adversarial Diffusion Distillation,” ECCV2024
> >
> > [Kang+2023] “Scaling up GANs for Text-to-Image Synthesis,” CVPR2023
> >
> > [Kang+2024] “Distilling Diffusion Models into Conditional GANs,” ECCV2024
> >
> > [Lin+2025] “Diffusion Adversarial Post-Training for One-Step Video Generation,” ICML2025
> >
> > [Xu+2025] “VideoGigaGAN: Towards Detail-rich Video Super-Resolution,” CVPR2025
> >
> > [Chen+2025] “NitroFusion: High-Fidelity Single-Step Diffusion through Dynamic Adversarial Training,” CVPR2025

---

> ### Author Response · Authors · 2025-11-21
> **Rebuttal Reply to Reviewer xeHQ (3/n)**
>
> ### [x-2] One of our main focuses in the experiment is to properly assess the effects of our method in general cases; the experiments based on GALIP demonstrate its effectiveness even in modern GANs
>
> > … the compared baselines are relatively outdated GAN architectures, making the reported improvements less meaningful.
>
> Thank you for your comment. We respectfully point out that our empirical results based on StudioGAN are insightful and that we also tested SONA using GALIP, which is a modern text-to-image GAN. Please find our detailed reply below.
>
> Our study focuses on conditional discriminators, and we have aimed to ensure fair comparisons by employing widely used architectures that are consistent across all baselines. To this end, we selected StudioGAN as our primary experimental environment, as it provides a well-curated benchmark that enables fair evaluation of a variety of conditional discriminators. We believe that the chosen GAN architecture backbones are appropriate for properly assessing the effects of the proposed methods.
>
> While recent strong class-conditional GANs such as StyleGAN-XL (Sauer+2022) utilize pre-trained vision foundation models in their discriminators to enhance generation performance as measured by FID scores, recent studies have reported that the use of such foundation models can compromise the integrity of evaluation metrics based on pre-trained models (Huang+2024). This phenomenon, often referred to as *feature leakage*, undermines essential comparisons. For this reason, we consider it necessary to use conventional GAN architectures in this work.
>
> Additionally, we would like to highlight our experiments based on GALIP, which is among the state-of-the-art GAN models for text-to-image generation. The improvements achieved by SONA in the experiments with GALIP indicate that SONA is also beneficial for modern GANs.
>
> [Sauer+2022] “StyleGAN-XL: Scaling StyleGAN to Large Diverse Datasets,” SIGGRAPH2022
>
> [Huang+2024] “The GAN is dead; long live the GAN! A Modern GAN Baseline,” NeurIPS2024
>
> ---
>
> ### [x-3] We have added contemporary generative models to our table for text-to-image to better position the performance of our method
>
> > The paper lacks baseline comparisons with modern diffusion models, which currently outperform GANs on most conditional generation benchmarks.
>
> While our scope is to develop a method that can be applicable to a broad range of discriminators instead of developing models outperforming diffusion models, we acknowledge the importance of positioning our method in relation to modern generative models. Therefore, we have **updated the table** reporting the results of text-to-image generation tasks under the zero-shot setting on COCO (**Table 5-B in the revised manuscript**). Please find our detailed reply below.
>
> We have included additional baselines beyond GANs (focusing on modern models trained on fewer than 1B data samples) in Table 5-B, situating SONA-based text-to-image models among these contemporary approaches. We would like to note that the comparison in Table 5-B is not strictly fair because GALIP is trained on the smallest dataset (the scale of the largest dataset among them is 860M$\gg$12M), as reported in the same table. Nevertheless, SONA applied to GALIP achieves competitive generation performance with the fastest inference speed among the listed models. Furthermore, because current text-to-image generative models such as SDXL and DALL-E 3 benefit from text–image datasets on a much larger scale, we believe that exploring this direction is an interesting future direction.

---

> > ### Author Response · Authors · 2025-11-21
> > **Rebuttal Reply to Reviewer xeHQ (4/4)**
> >
> > ### [x-4] We have quantitative results in Appendix G
> >
> > > Missing qualitative results. Given that the task involves image generation, it should include visual comparisons illustrating how the proposed method improves sample quality and conditional alignment relative to existing approaches.
> >
> > Thank you for your comment. In the initial manuscript, we presented qualitative results for SONA by including generated samples for all cases (see Figures 4, 5, 6, and 7). We also provided a comparison with the original GALIP trained on the CC12M dataset, which represents the most challenging task in our experimental setup, in Figure 7. For the other experimental setups, we intentionally omitted generated images from the baselines, as including all possible combinations would make the manuscript unnecessarily lengthy and potentially overwhelming for readers. To the best of our knowledge, most recent papers on generative models also focus on generated samples from their own models when presenting qualitative results (Sauer+2023, Huang+2024, Geng+2025), likely for the same reason.
> >
> > We would greatly appreciate it if you could specify which aspects you would like to see highlighted in the qualitative results. This feedback will help us better address your concerns in the revised manuscript.
> >
> > [Sauer+2023] “StyleGAN-T: Unlocking the Power of GANs for Fast Large-Scale Text-to-Image Synthesis,” ICML2023
> >
> > [Huang+2024] “The GAN is dead; long live the GAN! A Modern Baseline GAN,” NeurIPS2024
> >
> > [Geng+2025] “Mean Flows for One-step Generative Modeling,” NeurIPS2025

---

### Official Review · Reviewer_cYPq · 2025-11-03

**Soundness:** 3
**Presentation:** 3
**Contribution:** 3
**Rating:** 6
**Confidence:** 4

**Summary:**

This paper proposes SONA, a novel discriminator design for conditional Generative Adversarial Networks (GANs). It addresses the core challenge of balancing sample authenticity (naturalness) and conditional alignment by decomposing the discriminator's role into three integrated capabilities: robust unconditional discrimination, enhanced sensitivity to conditional alignment via negative samples, and dynamic adaptive weighting of these objectives. The method employs separate, orthogonal projections for naturalness and alignment, supported by a combination of Slicing Adversarial Network (SAN) and Bradley-Terry (BT) model-based losses. Extensive experiments on class-conditional (CIFAR-10, TinyImageNet, ImageNet) and text-to-image generation tasks demonstrate that SONA outperforms state-of-the-art methods in both sample quality and conditional alignment.

**Strengths:**

+ This paper proposes a novel discriminator architecture that explicitly decouples sample naturalness learning and conditional alignment through orthogonal projection. The combination of SAN and BT model objectives is innovative.

+ The paper is clearly written, explaining the proposed concepts in a simple and understandable way. The structure is logically rigorous, and the diagrams effectively complement the textual explanations.

+ The method validates its improvements in both standard and challenging benchmarks, and its applicability in text-to-image generation highlights its broad application potential.

**Weaknesses:**

- Lacks comparison with a simple dual-projection design, failing to demonstrate the necessity of the orthogonal constraint and potentially adding unnecessary complexity.
- Training speed is consistently slower than the simplest baseline (PD-GAN), without discussion of whether the performance gain justifies this overhead.
- The core theory relies on the strong, often unrealistic assumption of "perfect distribution matching," with no discussion of its validity in real-world training.
- The absence of a direct comparison with contemporary hybrid discriminators like ADC-GAN, which also integrate unconditional and conditional discrimination, makes it difficult to fully assess the innovative advantages.

**Questions:**

The paper shows that using a learnable ω_y for text conditioning degraded performance. Could the authors speculate more on the reasons for this? For instance, does the frozen CLIP embedding provide a more stable and semantically meaningful direction that is difficult to improve upon with a simple learned transformation? Have the authors explored more complex parameterizations (e.g., small MLPs) for ω_y?

Proposition 3 assumes p_g(x) = p_d(x). In practice, how sensitive is the effectiveness of the conditional alignment term V_BT-c to deviations from this perfect unconditional matching? Are there empirical observations from the experiments (e.g., from the MoG study) that suggest the conditional alignment learning remains effective even when unconditional FID is still being optimized?

The adaptive weighting mechanism is a simple solution. Did the authors experiment with or consider other adaptive weighting schemes from multi-task learning (e.g., based on uncertainty or task homoscedasticity) and, if so, how did they compare to the proposed method?

---

> ### Author Response · Authors · 2025-11-21
> **Rebuttal Reply to Reviewer cYPq (1/n)**
>
> Thank you for your constructive comments and your recognition of our work. Please kindly find our reply below.
>
> ---
>
> ### [c-1] Table 6 demonstrates the effectiveness of orthogonal projection
>
> > Lacks comparison with a simple dual-projection design, failing to demonstrate the necessity of the orthogonal constraint and potentially adding unnecessary complexity.
>
> Thank you for your comments. Our ablation study, reported in Table 6, demonstrates the effectiveness of the orthogonal projection. This table presents results for two scenarios: when the orthogonal projection is applied to SONA without the matching loss, the FID ($\downarrow$) is significantly improved (from 7.51 to 6.29); the projection also improves the FID when the matching loss is included (from 6.02 to 5.65). These results support the necessity of the orthogonal constraint.
>
> ---
>
> ### [c-2] Our new plots show the benefit of SONA compared to baselines in computational–performance trade-off
>
> > Training speed is consistently slower than the simplest baseline (PD-GAN), without discussion of whether the performance gain justifies this overhead.
>
> Thank you for your insightful comment regarding the trade-off between computational cost and performance. We agree that this is an important consideration. In response, we have **added a new subsection (Section 6.4)** that specifically addresses this issue. In particular, we have **included a new figure (Figure A)** in this section, which illustrates the benefits of SONA compared to PD-GAN. Please find our detailed response below.
>
> Although the training speed (specifically, training iterations per minute) was previously reported in Table 7 of the original manuscript (Appendix F.2.2), we have now included a new plot illustrating the relationship between **training time** and FID scores (see Figure A in Section 6.4 of the revised manuscript). This plot indicates that SONA achieves FID scores comparable to those of PD-GAN during approximately the first three days of training. After this period, SONA demonstrates a clear improvement, ultimately achieving lower FID scores. Notably, SONA attains a better FID score than the best result achieved by PD-GAN, and does so with a shorter training duration. Therefore, this new plot highlights the advantages of SONA over PD-GAN in terms of training efficiency.
>
> ---
>
> ### [c-3] Our new plots support that our conditional loss remains effective even when there is significant deviation in these distributions in practice, while this assumption facilitates theoretical development
>
> > The core theory relies on the strong, often unrealistic assumption of "perfect distribution matching," with no discussion of its validity in real-world training.
>
> Thank you for your careful review. We acknowledge that Proposition 3 relies on the strong assumption of unconditional distribution matching, which may not hold in practical situations. This assumption is imposed on the generator to facilitate theoretical development and to provide interpretable and clear insights. In the context of GANs, even stronger assumptions, such as discriminator optimality (e.g., Chu+2020), are often made for similar purposes because of the challenges associated with the theoretical analysis of practical GAN training. However, we provide empirical evidence below to demonstrate that our assumption is not required in practical cases.
>
> > Proposition 3 assumes p_g(x) = p_d(x). In practice, how sensitive is the effectiveness of the conditional alignment term V_BT-c to deviations from this perfect unconditional matching? Are there empirical observations from the experiments (e.g., from the MoG study) that suggest the conditional alignment learning remains effective even when unconditional FID is still being optimized?
>
> This is indeed a valid suggestion. It is important to consider whether such a strong assumption is required in practical scenarios. We have **updated Appendix E.2** to include empirical verification of this aspect (see **Figure B**), as explained below.
>
> To investigate this empirically, we have added a new figure that presents the curves of FID score (reflecting unconditional generation performance) and Top-1 accuracy (reflecting conditional alignment) with respect to training steps in the ImageNet case presented in Section 6.1 (see Figure B in Appendix E.2). This visualization shows that both metrics improve simultaneously as training progresses in SONA as well as in the other two methods. These results suggest that conditional learning based on $\mathcal{V}_{\text{BT-c}}$ remains effective even when the unconditional FID is still high, indicating that $p_g(x) = p_d(x)$ does not yet hold.
>
> [Chu+2020] “Smoothness and Stability in GANs,” ICLR2020

---

> ### Author Response · Authors · 2025-11-21
> **Rebuttal Reply to Reviewer cYPq (2/n)**
>
> ### [c-4] The classifier-based baselines are based on hybrid discriminators, and ReACGAN is state-of-the-art in this category on a well-curated benchmark
>
> > The absence of a direct comparison with contemporary hybrid discriminators like ADC-GAN, which also integrate unconditional and conditional discrimination, makes it difficult to fully assess the innovative advantages.
>
> Thank you for suggesting ADC-GAN. Please find our response below, which addresses two main aspects.
>
> **(1) Contemporary hybrid discriminators, including ADC-GAN, are categorized into the classifier-based approach discussed in this paper.**
>
> As described in Section 2.2, classifier-based methods refer to GANs with discriminators that produce outputs for both unconditional discrimination and conditional alignment. We selected ContraGAN and ReACGAN as representative methods of this approach, which have demonstrated strong performance in previous studies (Kang+2023). It is also important to note that the classifier-based approach has not been adopted in current text-to-image GANs such as GALIP, StyleGAN-T, and GigaGAN.
>
> **(2) ReACGAN achieves best performance among the classifier-based methods, while ADCGAN performs inferior to the other baselines in our setups.**
>
> We intentionally excluded ADC-GAN from our main comparison table because, in our preliminary experiments, we found that its training was less stable than that of other classifier-based methods, resulting in scores that were significantly worse than those reported in the original paper when trained using the StudioGAN repository. For reference, we report the performance of ADC-GAN on TinyImageNet (with and without DiffAug, following Table 3) and ImageNet (batch size 256) below. These results show that ReACGAN achieves the best values on many metrics among the classifier-based (hybrid discriminator) methods.
>
> **TinyImageNet**
> | Method | FID $\downarrow$ | IS $\uparrow$ | Dens $\uparrow$ | Cover $\uparrow$ |
> | ---- | ---- | ---- | ---- | ---- |
> | **ADC-GAN** | 28.21$\pm$1.11 | 13.50$\pm$0.29 | 0.57$\pm$0.03 | 0.48$\pm$0.01 |
> | ContraGAN | 23.66$\pm$1.59 | 12.47$\pm$0.45 | 0.62$\pm$0.05 | 0.46$\pm$0.03 |
> | ReACGAN | 18.99$\pm$0.98 | 15.37$\pm$0.68 | 0.70$\pm$0.03 | 0.54$\pm$0.02 |
> | PD-GAN | 20.77$\pm$1.53 | 14.29$\pm$1.11 | 0.70$\pm$0.05 | 0.58$\pm$0.02 |
> | SONA | 16.33$\pm$0.62 | 16.60$\pm$0.35 | 0.74$\pm$0.02 | 0.59$\pm$0.01 |
>
> **TinyImageNet (w/ DiffAug)**
> | Method | FID $\downarrow$ | IS $\uparrow$ | Dens $\uparrow$ | Cover $\uparrow$ |
> | ---- | ---- | ---- | ---- | ---- |
> | **ADC-GAN** | 16.72$\pm$0.49 | 16.22$\pm$0.16 | 0.61$\pm$0.06 | 0.63$\pm$0.01 |
> | ContraGAN | 11.86$\pm$0.32 | 16.01$\pm$0.29 | 0.78$\pm$0.02 | 0.63$\pm$0.01 |
> | ReACGAN | 9.93$\pm$0.34 | 20.25$\pm$0.07 | 0.88$\pm$0.01 | 0.69$\pm$0.00 |
> | PD-GAN | 13.09$\pm$1.00 | 16.57$\pm$0.34 | 0.78$\pm$0.02 | 0.70$\pm$0.02 |
> | SONA | 7.76$\pm$0.29 | 23.00$\pm$0.10 | 0.99$\pm$0.01 | 0.79$\pm$0.00 |
>
> **ImageNet-128 (batch size 256)**
> | Method | FID $\downarrow$ | IS $\uparrow$ | Dens $\uparrow$ | Cover $\uparrow$ | Top1/5 acc $\uparrow$ |
> | ---- | ---- | ---- | ---- | ---- | ---- |
> | **ADC-GAN** | 21.71 | 40.44 | 0.57 | 0.49 | 0.38/0.61 |
> | ContraGAN | 31.73 | 23.93 | 0.57 | 0.28 | 0.02/0.09 |
> | ReACGAN | 18.73 | 51.29 | 0.85 | 0.46 | 0.20/0.48 |
> | PD-GAN | 29.76 | 27.17 | 0.45 | 0.35 | 0.24/0.48 |
> | SONA | 13.17 | 83.33 | 0.79 | 0.59 | 0.62/0.87 |
>
> We would like to note that ADC-GAN is already implemented in the StudioGAN repository, and we used this implementation for our experiments. Additionally, we did not find any significant deviation between the implementation and the original paper.
>
> If you believe that reporting the results of ADC-GAN would be beneficial for readers, we are happy to include its results for all metrics in Table 3. Additionally, we are currently training ADC-GAN on ImageNet with the batch size 2048 as well, and expect these experiments to be completed by December 2nd.
>
> [Kang+2023] “StudioGAN: A Taxonomy and Benchmark of GANs for Image Synthesis,” TPAMI

---

> > ### Author Response · Authors · 2025-11-21
> > **Rebuttal Reply to Reviewer cYPq (3/3)**
> >
> > ### [c-5] We suspect that maintaining CLIP embeddings pre-trained on large-scale text–image pairs is beneficial for improving text alignment
> >
> > > The paper shows that using a learnable ω_y for text conditioning degraded performance. Could the authors speculate more on the reasons for this? For instance, does the frozen CLIP embedding provide a more stable and semantically meaningful direction that is difficult to improve upon with a simple learned transformation? Have the authors explored more complex parameterizations (e.g., small MLPs) for ω_y?
> >
> > Thank you for your constructive suggestion. We agree that discussing the potential reasons for this issue will be helpful for readers. We suspect that the underlying reason is that a learnable module for outputting $\omega$ may either retain most of the information from the CLIP embeddings or lose a portion of it, as suggested by information theory. CLIP was pre-trained on a much larger scale of text–image data compared with datasets such as CUB, COCO, and CC12M, which allows the CLIP embeddings to contain rich information that is effective even in zero-shot scenarios. In contrast, when a learnable module is introduced to connect the CLIP embeddings to $\omega_y$, this module is trained only on smaller-scale datasets. This limitation may prevent the module from fully leveraging the effective information obtained through pre-training and may result in overfitting to the training dataset. Therefore, retaining the CLIP embedding without any transformation appears to be the most effective strategy for achieving text–image alignment.
> >
> > While we explored several approaches to apply a learnable $\omega$ to SONA in the GALIP case (for example, using a single affine layer or a three-layer MLP to output $\omega_y$ from the CLIP embeddings), these methods did not lead to improved generation performance; in fact, we observed either deterioration or no improvement.
> >
> > ---
> >
> > ### [c-6] Our new experiment comparing our adaptive weighting with an existing method highlights the benefit of our approach
> >
> > > The adaptive weighting mechanism is a simple solution. Did the authors experiment with or consider other adaptive weighting schemes from multi-task learning (e.g., based on uncertainty or task homoscedasticity) and, if so, how did they compare to the proposed method?
> >
> > Thank you for your valuable suggestion. We agree that comparing our adaptive weighting approach with existing methods is an effective way to highlight our contribution. In response, we have **added new empirical results** demonstrating that our method is effective compared with one of the most basic adaptive weighting methods from multi-task learning (Kendall et al., 2018), to **Section 4.4 in the revision**. Additionally, we have added the discussion from the last paragraph of Appendix E.4 to Section 4.4 to further enhance the visibility of the motivation of our adaptive weighting method. We believe these revisions help to highlight our contribution. Please find our detailed response below.
> >
> > We trained SONA using the adaptive weighting method proposed by Kendall et al. (2018) under the experimental setup described in Section 6.3. The results are presented below, following the format of Table 6.
> >
> > | Weighting coefficients | FID $\downarrow$ | IS $\uparrow$ |
> > | ---- | ---- | ---- |
> > | Fixed coefficients | 7.09$\pm$1.17 | 9.52$\pm$0.07 |
> > | Kendal et al. | 16.62$\pm$4.04 | 7.88$\pm$0.80 |
> > | Our adaptive weighting | 5.65$\pm$0.25 | 9.51$\pm$0.05 |
> >
> > These results further support the motivation for developing our simple adaptive weighting scheme. As briefly discussed in the last paragraph of Appendix E.4 (in the initial manuscript), the most basic method from multi-task learning balances multiple objective terms using **unbounded** coefficients. We suspect that the unbounded nature of these coefficients is a critical factor that can cause instability in GAN training, since GAN training is well known to be highly sensitive to the learning rate setting, although such methods are effective in a broad range of machine learning applications. Based on this hypothesis, we proposed a method that balances multiple discriminator objectives using **bounded** coefficients, which are empirically shown to be effective.
> >
> > [Kendall+2018] “Multi-task learning using uncertainty to weigh losses for scene geometry and semantics.” CVPR2018.

---

> ### Author Response · Authors · 2025-11-26
> **Additional experimental results of  ADC-GAN on ImageNet-128**
>
> Thank you for taking the time to review our paper again.
>
> As promised in our initial reply ([c-4]), we would like to report the generation performance of ADC-GAN trained on ImageNet-128 with a batch size of 2048 below. The result is almost consistent with those in our initial reply: ReACGAN achieves superior performance to ADC-GAN in terms of major metrics such as FID and IS, while SONA outperforms all the baselines including them.
>
> **ImageNet (batch size=2048)**
> | Method | FID $\downarrow$ | IS $\uparrow$ | Dens $\uparrow$ | Cover $\uparrow$ | Top1/5 acc $\uparrow$ |
> | ---- | ---- | ---- | ---- | ---- | ---- |
> | **ADC-GAN** | 9.75 | 98.80 | 0.79 | 0.78 | 0.65/0.84 |
> | ReACGAN | 8.44 | 103.07 | 1.04 | 0.71 | 0.51/0.82 |
> | PD-GAN | 8.85 | 96.11 | 0.95 | 0.81 | 0.63/0.83 |
> | SONA | 6.14 | 140.14 | 1.03 | 0.82 | 0.80/0.93 |
>
>
> We hope we have addressed all your comments. If you have any further questions or concerns, we would appreciate your feedback. Thank you again for your time and consideration.

---

### Official Review · Reviewer_ip5t · 2025-11-05

**Soundness:** 3
**Presentation:** 3
**Contribution:** 3
**Rating:** 6
**Confidence:** 3

**Summary:**

The paper proposes SONA, a conditional GAN discriminator that focuses on these two objectives (a) unconditional “naturalness” scoring from (b) conditional alignment via orthogonal projections, and augments training with mismatching-aware supervision and adaptive weighting across objectives.
The unconditional term is trained with a slicing adversarial networks (SAN) objective, while alignment uses two Bradley–Terry pairwise losses—one for conditional discrimination and one for mismatch awareness—plus a light adaptive weighting to keep the objectives balanced.
The method’s overall loss learns data realism and condition faithfulness without the two interfering. Empirically, SONA improves FID/IS across CIFAR-10, TinyImageNet, and ImageNet-128; it also plugs into a text-to-image GAN (GALIP) and gets consistent FID gains The paper includes propositions analyzing the objectives and ablations supporting the design choices.

**Strengths:**

- The proposal of “sum of naturalness + alignment” with an explicit orthogonal projection is a nice inductive bias and factorization that reduces interference between realism and conditioning by construction in the objective function.
- The paper shows consistent improvement across standard cGAN settings, including a nice ImageNet-128 table and a clean plug-in to GALIP for text-to-image.
- The paper shows the ablation experiments that separates out the impact of the orthogonal split, mismatch-aware loss, and adaptive weights.
- The source code is attached in the supplementary for reproducibility.

**Weaknesses:**

- The experimental comparisons emphasize AC-GAN-style and projection GANs; some more discussion on strong GAN baselines and competitive diffusion/rectified-flow generators could help strengthen the empirical case, especially on ImageNet.
- The paper introduces additional pairwise losses and learnable weighting under a normalization constraint, maybe some discussion on the compute/memory overhead could help readers evaluate the efficacy of the proposed approach reproduce gains reliably.
- For GALIP, ω_y is frozen from CLIP and the CLIP-score improvement seems kind of incremental. It could help the readers if larger-scale text-to-image (higher res, varied prompts) and a learnable ω_y variant experiment could be performed to check the alignment benefits in richer/diverse settings.

**Questions:**

should there be :

Consistent usage of matching-aware vs mismatching-aware throughout the paper. In Abstract (and consistently across the paper): “matching-aware (supervision/discrimination)” → “mismatching-aware …” (to match the title and the negative-pair supervision described).

---

> ### Author Response · Authors · 2025-11-21
> **Rebuttal Reply to Reviewer ip5t (1/n)**
>
> Thank you for your feedback and positive comments, which encourages us. Please kindly find our reply below.
>
> ---
>
> ### [i-1] We have added contemporary generative models to our table for text-to-image to better position the performance of our method
>
> > The experimental comparisons emphasize AC-GAN-style and projection GANs; some more discussion on strong GAN baselines and competitive diffusion/rectified-flow generators could help strengthen the empirical case, especially on ImageNet.
>
> Thank you for your valuable comments. Your suggestion is helpful for better positioning our method in relation to modern generative models. In response, we have **updated the table** reporting the results of text-to-image generation tasks under the zero-shot setting on COCO (**Table 5-B in the revised manuscript**). Please find our detailed reply below.
>
> Our study focuses on conditional discriminators, and we aim to ensure fair comparisons by employing commonly used architectures that are shared across all baselines. For this reason, we selected StudioGAN as our primary experimental environment, as it is a well-curated benchmark that facilitates fair evaluation of a wide range of conditional discriminators. Furthermore, most conditional discriminators used in previous literature adopt either a classifier-based or a projection-based approach. Nevertheless, we recognize that discussing SONA in the context of current strong baselines would be beneficial for readers. We address this aspect through experiments on GALIP, as described below.
>
> GALIP serves as a **strong baseline** despite its efficient training and lightweight architectural backbone. Since its introduction in 2023, GALIP has remained comparable to state-of-the-art text-to-image generative models on CC12M and datasets of similar scale, even when compared with diffusion- or flow-based models. To provide a numerical context, we have included additional baselines (focusing on modern models trained on fewer than 1B data samples) in Table 5-B, situating SONA-based text-to-image models among these contemporary approaches. We would like to note that the comparison in Table 5-B is not strictly fair because GALIP is trained on the smallest dataset (the scale of the largest dataset among them is 860M$\gg$12M), as reported in the same table. Nevertheless, SONA applied to GALIP achieves competitive generation performance with the fastest inference speed among the listed models. Furthermore, because current text-to-image generative models such as SDXL and DALL-E 3 benefit from text–image datasets on a much larger scale, we believe that exploring this direction is an interesting future direction.
>
> Given that text-to-image generation tasks on CC12M are more challenging than class-conditional generation on ImageNet-128, and that current generation performance on ImageNet is nearly saturated, we believe that our experiments on GALIP provide a more effective demonstration of the competitiveness of our approach.

---

> ### Author Response · Authors · 2025-11-21
> **Rebuttal Reply to Reviewer ip5t (2/n)**
>
> ### [i-2] Our new plots show the benefit of SONA compared to baselines in computational–performance trade-off, despite marginal memory overhead
>
> > … some discussion on the compute/memory overhead could help readers evaluate the efficacy of the proposed approach reproduce gains reliably.
>
> Thank you for your encouraging comment. In response, we have **added a new figure** to further discuss our method in terms of compute overhead in the revised manuscript (see **Figure A in Section 6.4**). Please also refer to Table 7 in Appendix F.2.2. Our detailed response follows below.
>
> Regarding the **compute overhead**, we reported training iterations per minute in Table 7 of the initial manuscript. This table shows that SONA is slower than PD-GAN in training time per iteration, while it is faster than classifier-based methods such as ContraGAN and ReACGAN.
>
> To further address your comment, we have included a new plot illustrating the relationship between **training time** and FID scores based on experiments on ImageNet with a batch size of 2048 (Figure A). This plot helps compare SONA with the baselines from the perspective of computational–performance trade-off. The results indicate that SONA achieves FID scores comparable to those of PD-GAN during approximately the first three days of training. After this period, SONA demonstrates a clear improvement, ultimately achieving lower FID scores. Notably, SONA attains a better FID score than the best result achieved by PD-GAN, and does so with a shorter training duration. Therefore, this new plot highlights the advantages of SONA over PD-GAN in terms of training efficiency.
>
> As for the **memory overhead / parameter size**, SONA maintains almost the same number of model parameters as PD-GAN, and therefore requires only minimal additional memory. The architectural differences between SONA and PD-GAN are listed below:
> - In SONA, three learnable scalar parameters are added on top of PD-GAN, which are used to balance the objective terms included in the discriminator loss. The additional memory usage introduced by this is negligible.
> - While the unconditional linear projection and conditional projection are normalized in their norms in SONA, this normalization does not introduce any additional parameters.  The additional memory usage introduced by this is negligible.
> - While the last feature in SONA’s discriminator is processed by the orthogonal projection, this operator does not introduce any additional parameters. The additional memory usage introduced by this is negligible.

---

> ### Author Response · Authors · 2025-11-21
> **Rebuttal Reply to Reviewer ip5t (3/3)**
>
> ### [i-3] We suspect that maintaining CLIP embeddings pre-trained on large-scale text–image pairs is beneficial for improving text alignment
>
> > …It could help the readers if larger-scale text-to-image (higher res, varied prompts) and a learnable ω_y variant experiment could be performed to check the alignment benefits in richer/diverse settings.
>
> While we explored several approaches to apply a learnable $\omega$ to SONA in the GALIP case (for example, using a single affine layer or a three-layer MLP to output $\omega_y$ from the CLIP embeddings), these methods did not lead to improved generation performance; in fact, we observed either deterioration or no improvement. Since conducting larger-scale text-to-image experiments is challenging, as described later, we would like to elaborate on the potential reason for this issue below.
>
> We suspect that the underlying reason is that a learnable module for outputting $\omega$ may either retain most of the information from the CLIP embeddings or lose a portion of it, as suggested by information theory. CLIP was pre-trained on a much larger scale of text–image data compared with datasets such as CUB, COCO, and CC12M, which allows the CLIP embeddings to contain rich information that is effective even in zero-shot scenarios. In contrast, when a learnable module is introduced to connect the CLIP embeddings to $\omega_y$, this module is trained only on smaller-scale datasets. This limitation may prevent the module from fully leveraging the effective information obtained through pre-training and may result in overfitting to the training dataset. Therefore, retaining the CLIP embedding without any transformation appears to be the most effective strategy for achieving text–image alignment.
>
> Lastly, we would like to note that, to the best of our knowledge, GALIP is the only model that is publicly available and **reproducible**. StyleGAN-T could be another candidate for testing SONA in a larger-scale text-to-image generation setup. However, we found that the official code does not reproduce the results reported in their paper, which is also reported in the “Issues” section of the repository.
>
> > For GALIP, ω_y is frozen from CLIP and the CLIP-score improvement seems kind of incremental.
>
> We would like to emphasize that the CLIP scores achieved by GALIP (for example, 0.3411 on the COCO dataset) are already competitive or higher compared to those of current state-of-the-art models, including diffusion-based approaches that are trained on much larger datasets. For reference, please see the CLIP scores on the COCO dataset reported in recent text-to-image generative models, such as Figure 5 in [Sauer et al., 2023], Table 3 in [Kang et al., 2023], Figure 12 in [Podell et al., 2024], and Table 1 in [Batker+]. Nevertheless, we observe that SONA provides a modest improvement in CLIP score, while leading to a significant improvement in FID.
>
> [Sauer+2023] “StyleGAN-T: Unlocking the Power of GANs for Fast Large-Scale Text-to-Image Synthesis,” ICML2023
>
> [Kang+2023] “Scaling up GANs for Text-to-Image Synthesis,” CVPR2023
>
> [Podell+2024] “SDXL: Improving Latent Diffusion Models for High-Resolution Image Synthesis,” ICLR2024
>
> [Batker+] “Improving Image Generation with Better Captions,” technical report of DALL·E 3
>
> ---
>
> ### [i-4] We have revised the manuscript to consistently use “matching-aware”
>
> > Consistent usage of matching-aware vs mismatching-aware throughout the paper.
>
> Thank you for bringing this issue to our attention. We have revised the manuscript to ensure consistent usage of “matching-aware” throughout, rather than “mismatching-aware.”

---

### Author Response · Authors · 2025-11-21
**Message to the reviewers**

Dear Reviewers,

Thank you very much for your time and effort in reviewing our paper and for your insightful feedback. We are pleased that the reviewers found our manuscript to be well-organized and clearly written in a simple and understandable manner (cYPq, xeHQ). We appreciate your recognition of our technical contributions (ip5t, cYPq, nbxK) and theoretical development (nbxK). We are also grateful that you acknowledged the comprehensiveness of our experiments and recognized that our method is broadly applicable for improving conditional discriminators (ip5t, cYPq, xeHQ, nbxK).

We have carefully revised our manuscript to address your comments and suggestions. The updated manuscript includes the following changes, which are highlighted in blue:

- **Title (ip5t):** We have changed the term “mismatching-aware” to “matching-aware.”
  - We have made similar modifications throughout the manuscript to consistently use “matching-aware” instead of “mismatching-aware.”
  - We have also revised the manuscript to consistently use “matching loss” instead of “mismatching loss.”
  - We have additionally revised some notations accordingly (e.g., $\mathcal{L}\_{\text{BT-MM}}\Longrightarrow\mathcal{L}\_{\text{BT-M}}$) throughout the manuscript.
- **Section 4.4 (cYPq):** We have added a paragraph on our adaptive weighting mechanism.
  - We have elaborated further on the motivation for our adaptive weighting mechanism.
  - We have also empirically compared our method with a basic baseline from the multi-task learning context to highlight the benefits of our approach.
- **Section 6.2 (ip5t, xeHQ, nbxK):** We have updated the table reporting the results of text-to-image generation tasks under the zero-shot setting on COCO, which is temporarily labeled as **Table 5-B**,
  - We have included additional baselines beyond GANs to better situate SONA-based text-to-image models among contemporary approaches.
  - We have also added some sentences to Section 6.2 accordingly.
- **Section 6.4 (ip5t, cYPq, nbxK):** We have added Section 6.4 to provide a more detailed discussion of computational time.
  - We have moved Table 7 from Appendix F.2.2 to Section 6.4, where it is temporarily labeled as **Table A**, to enhance accessibility.
  - We have included **Figure A** to demonstrate that SONA is more efficient than the baselines in terms of the computational–performance trade-off.
- **Appendix A (nbxK):** We have added sentences to highlight the computational efficiency of SONA compared with the classifier-based approach.
- **Appendix B (nbxK):** We have expanded this section by adding dedicated subsections on the sliced Wasserstein and the Bradley-Terry framework to assist readers who may not be familiar with these concepts. We have references to these subsections in the body accordingly.
- **Appendix E.2 (cYPq):** We have added **Figure B** to show that both metrics improve simultaneously as training progresses in SONA on the ImageNet case, providing empirical evidence that the distribution matching assumption made in Proposition 3 is not necessary in practical cases.
- **Appendix H (xeHQ, nbxK):** We have added a paragraph to discuss potential future work beyond GANs.

We are grateful for your constructive feedback, which has significantly helped us to refine and strengthen our paper. Currently, we are using alphabetic labels for additional figures and tables to avoid changes from the initial submission. We will renumber all figures and tables in the camera-ready version.

---

> ### Author Response · Authors · 2025-12-01
> **Update regarding the general comment**
>
> We have completed the experiments on ADC-GAN as suggested by Reviewer cYPq and have reported the additional results in the discussion thread.
>
> We believe that we have addressed all questions and concerns raised by the reviewers and have incorporated their feedback into the current manuscript to enhance its quality.

---

### Author Response · Authors · 2025-11-26
**A kind reminder for further discussion**

Dear Reviewers,

Thank you for dedicating your time to review our paper once again. There is currently about one week remaining in the reviewer--author discussion period. We look forward to receiving your further feedback.

---

### Author Response · Authors · 2025-12-02
**Rebuttal summary to ACs**

Dear (new) Area Chairs,

Thank you for your efforts in coordinating our submission under the unusual circumstances. Although we submitted our responses on **Nov. 20**, the discussion period closed before we received reviewer feedback. To facilitate your understanding, we provide a brief summary to cover all of our responses below. Please refer to **Message to the reviewers** and the full responses for details.

---
## [Part I] Discussion points raised by multiple reviewers

**[C1] Computational complexity** (corresponding to our replies **i-2**, **c-2**, and **n-2**)

Table 7 in our initial manuscript shows that SONA requires more time per **iteration** than the simplest method, PD-GAN. However, our new plots (**Fig. A**) demonstrate that SONA is the most computationally efficient in terms of generation performance relative to **training time**. We have added **Sec. 6.4** to discuss this computational perspective.

---

**[C2] Position of this work in relation to modern generative models beyond GANs** (corresponding to our replies **i-1**, **x-1**, and **n-3**)

We have clarified this aspect in two ways: (1) added modern baselines beyond GANs in the text-to-image (T2I) experiment (**Tab. 5-B**) to better situate SONA-based models among contemporary approaches; (2) expanded the future work section to discuss the potential applicability of SONA to diffusion distillation (**Appx. H**).

---

**[C3] Discussion on why learnable $\omega_y$ can degrade the performance in T2I** (corresponding to our replies **i-3** and **c-5**)

As reported in the initial manuscript, several straight approaches to applying a learnable $\omega_y$ to SONA in the GALIP case did not improve generation performance compared with frozen CLIP embedding. We suspect that, though CLIP embeddings pre-trained on large-scale text-image pairs contain rich information that is effective even in zero-shot scenarios, training $\omega_y$ on a smaller-scale dataset (such as COCO or CC12M) may lose a portion of it.

---

> ### Author Response · Authors · 2025-12-02
> **(Continued) Rebuttal summary to ACs**
>
> ## [Part II] Our responses to individual comments
>
> **Comments from Reviewer ip5t:** three weaknesses (W1-3) and one question (Q)
> - **W1:** More discussion on recent generative models within and beyond GANs.
>   - **[Ans] i-1:** Beyond GANs, we have added more baselines to **Tab. 5-B** (see **[C2]** above). Within GANs, GALIP serves as a strong recent baseline.
> - **W2:** Justification of computational/memory overhead.
>   - **[Ans] i-2:** We have added **Fig. A** for the computational aspect (see **[C1]** above). Regarding memory, SONA maintains almost the same model parameter size as PD-GAN.
> - **W3:** For GALIP, $\omega_y$ is frozen from CLIP, and CLIP-score improvement is marginal
>   - **[Ans] i-3:** Please refer to **[C3]** above for our intuitions for frozen CLIP. Regarding the CLIP scores, we emphasize that those achieved by GALIP are already competitive or higher compared to those of current state-of-the-art models.
> - **Q:** Mixed usage of matching-aware vs mismatching-aware throughout the paper.
>   - **[Ans] i-4** Revised manuscript to consistently use “matching-aware.”
>
> ---
>
> **Comments from Reviewer cYPq:** four weaknesses (W1-4) and three questions (Q1-3)
> - **W1:** Lacks of experiments to show the necessity of the orthogonal constraint.
>   - **[Ans] c-1:** No, Tab. 6 demonstrates its effectiveness.
> - **W2:** Justification of computational overhead.
>   - **[Ans] c-2:** We have added **Fig. A** to show the training efficiency of SONA (see **[C1]** above).
> - **W3&Q2:** Discussion on the distributional matching $p_g(x) = p_d(x)$ assumed in Prop. 3.
>   - **[Ans] c-3:** New plots (**Fig. B**) confirmed that this assumption is not necessary in practice, while it is useful for theory development.
> - **W4:** Absence of a direct comparison with contemporary hybrid discriminators like ADC-GAN.
>   - **[Ans] c-4:** No, *Classifier-based discriminators* referenced in our paper (e.g., ContraGAN, ReACGAN) are identical to *hybrid discriminators*. Additionally, we have newly trained ADC-GAN on all class-conditional cases.
> - **Q1:**  Why can using a learnable $\omega_y$ for text conditioning degrade performance?
>   - **[Ans] c-5:** Please refer to **[C3]** above for our hypothesis.
> - **Q3:** Do other adaptive weighting schemes from multi-task learning work?
>   - **[Ans] c-6:** New experiment added to **Sec. 4.4** shows that training SONA with a method from this line resulted in much worse generation performance.
>
> ---
>
> **Comments from Reviewer xeHQ:** three weaknesses (W1-3)
> - **W1.1:** Benefit of this research in the current deep generative modeling.
>   - **[Ans] x-1:** Since our method is applicable to **general conditional discriminators** in principle, it has the potential to improve diffusion distillation (see **[C2]** above) as well as a broad range of GANs, which offer unique benefits such as fast sampling compared with diffusion models.
> - **W1.2:** The reported improvements are less meaningful due to outdated GAN architectures.
>   - **[Ans] x-2:** We used StudioGAN for class-conditional cases, which we believe is the best benchmark for fair comparisons. Additionally, SONA improves the performance even in GALIP, a modern GAN.
> - **W2:** Lacks of baseline comparisons with diffusion models.
>   - **[Ans] x-3:** We have added more baselines to **Tab. 5-B** (see **[C2]** above).
> - **W3:** Missing qualitative results.
>   - **[Ans] x-4:** No, quantitative results are included in Appendix G.
>
> ---
>
> **Comments from Reviewer nbxK:** four weaknesses (W1-4) and four questions (Q1-4)
> - **W1.1:** SONA may make training more cumbersome.
>   - **[Ans] n-1:** SONA is simpler than classifier-based approaches in practice.
> - **W1.2&Q1&Q2:** More discussion on computational overhead and convergence behavior.
>   - **[Ans] n-2:** We have added **Fig. A** for the computational aspect (see **[C1]** above). **Fig. A** also demonstrates that SONA training stably converges in practice.
> - **W2&Q3:** Limited comparison with diffusion-based models.
>   - **[Ans] n-3:** We have added more baselines beyond GANs to **Tab. 5-B** (see **[C2]** above).
> - **W3.1:** Ablation study on hyperparameter selection.
>   - **[Ans] n-4:** SONA does not include any hyperparameters beyond those for unconditional GANs.
> - **W3.2:** Discussion on limitations.
>   - **[Ans] n-5:** Appx. H describes potential limitations from various perspectives.
> - **W4:** Clarity and accessibility regarding prior knowledge.
>   - **[Ans] n-6:** We have expanded Appx. B by adding dedicated subsections introducing SAN and the Bradley-Terry framework.
> - **Q4:** Scaling to more complex condition spaces than discrete labels.
>   - **[Ans] n-7:** CLIP embedding, used in GALIP for T2I, is continuous and high-dimensional.

---

### Meta-Review · Area_Chair_VRGT · 2025-12-28

**Summary:**

The paper received borderline recommendations that slightly leaned toward acceptance (three weak accepts and one weak reject). The main concerns raised by the reviewers were: (1) the relevance and significance of the contribution in the context of modern generative modeling dominated by diffusion/flow models, (2) computational overhead and training efficiency, (3) strong theoretical assumptions (e.g., perfect marginal matching) and their practical implications, and (4) missing or insufficient baseline comparisons and ablation studies.

**Reviewer Concerns:**

After examining the revised manuscript and rebuttal, the AC finds that the concerns about computational overhead, baseline/ablation coverage, and the practical relevance of the theoretical assumptions were reasonably addressed. In particular, the authors added additional baseline results (including contemporary hybrid discriminator comparisons) and provided a more informative analysis of the compute–performance trade-off, alongside empirical evidence indicating that the key mechanism remains effective even when the strict theoretical assumptions do not hold during training.

The remaining concern, primarily raised by reviewer xeHQ, is that the overall impact may be limited given the field’s shift toward diffusion-based generators. This concern is only partially mitigated: while the authors improved the positioning and discussion, the paper does not aim to compete directly with diffusion models and focuses on discriminator design within adversarial training. Nevertheless, under this stated scope, the proposed discriminator framework is technically well-motivated and empirically validated, and the AC does not view the remaining impact-related concern as sufficient grounds for rejection.

**Reviewer Scores:**

Based on the rebuttal, the positive reviewers are likely to maintain their recommendations. The negative reviewer is likely to maintain borderline recommendations.

---

### Decision · Program_Chairs · 2026-01-26

Accept (Poster)